# Ultrafast nematic-orbital excitation in FeSe

T. Shimojima[1,2], Y. Suzuki[2], A. Nakamura[1,2], N. Mitsuishi[2], S. Kasahara [3], T. Shibauchi [4], Y. Matsuda [3], Y. Ishida[5], S. Shin [5] & K. Ishizaka[1,2]

The electronic nematic phase is an unconventional state of matter that spontaneously breaks the rotational symmetry of electrons. In iron-pnictides/chalcogenides and cuprates, the nematic ordering and fluctuations have been suggested to have as-yet-unconfirmed roles in superconductivity. However, most studies have been conducted in thermal equilibrium, where the dynamical property and excitation can be masked by the coupling with the lattice. Here we use femtosecond optical pulse to perturb the electronic nematic order in FeSe. Through time-, energy-, momentum- and orbital-resolved photo-emission spectroscopy, we detect the ultrafast dynamics of electronic nematicity. In the strong-excitation regime, through the observation of Fermi surface anisotropy, we find a quick disappearance of the nematicity followed by a heavily-damped oscillation. This short-life nematicity oscillation is seemingly related to the imbalance of Fe $3d_{xz}$ and $d_{yz}$ orbitals. These phenomena show critical behavior as a function of pump fluence. Our real-time observations reveal the nature of the electronic nematic excitation instantly decoupled from the underlying lattice.

[1] RIKEN Center for Emergent Matter Science (CEMS), Wako 351-0198, Japan. [2] Quantum-Phase Electronics Center (QPEC) and Department of Applied Physics, The University of Tokyo, Tokyo 113-8656, Japan. [3] Department of Physics, Kyoto University, Kyoto 606-8502, Japan. [4] Department of Advanced Materials Science, The University of Tokyo, Kashiwa 277-8561, Japan. [5] Institute for Solid State Physics (ISSP), The University of Tokyo, Kashiwa 277-8581, Japan. Correspondence and requests for materials should be addressed to T.S. (email: takahiro.shimojima@riken.jp)

Iron-based superconductors exhibit attractive properties such as high-transition-temperature ($T_c$) superconductivity and complex competing phases[1]. Their electronic structures consist of multiple iron $3d$ orbitals, thus giving rise to a variety of antiferroic and ferroic ordering phenomena involving spin and orbital profiles[2–4]. Among these mysterious phases, there has been increasing interest in nematic order[5–13], which spontaneously breaks the rotational symmetry of electrons and triggers a lattice instability[8]. Recent investigations by electronic Raman scattering[14,15] and elastoresistivity measurements[11] unveiled the fluctuation of electronic nematicity and its critical behavior commonly in the optimally doped regimes of different material families[11].

FeSe exhibits superconductivity ($T_c = 9$ K) and a nematic order accompanied by the tetragonal-to-orthorhombic lattice deformation ($T_s = 90$ K) without any magnetic order[16]. Its electronic structure is given in Fig. 1 (see also Supplementary Note 1). In the tetragonal phase, FeSe exhibits a circular Fermi surface around the $\Gamma$ point (Fig. 1a). Along $k_x$ ($k_y$), the hole band forming the Fermi surface has the $yz$ ($xz$) orbital component. Note that such

momentum ($k$)-dependent orbital characters keep the four-fold ($C_4$) symmetry (Fig. 1b). In the orthorhombic (nematic) phase, the $k$-dependent orbital polarization[17] modifies the Fermi surface shape into an elliptical one (nematic Fermi surface) as shown in Fig. 1c, resulting in inequivalent Fermi momenta ($k_F$) along $k_x$ and $k_y$ ($k_{Fx} < k_{Fy}$). At the same time, the orbital components at the $k_F$'s are mixed, especially along $k_x$ as shown in Fig. 1d (Supplementary Note 2). While these characteristics associated with the nematic order have been verified through intensive angle-resolved photoemission spectroscopy (ARPES) studies[17–20], the understanding of dynamics and excitations peculiar to this condensed state remain lacking.

Time-resolved ARPES (TARPES) has the potential impact to resolve electron dynamics not only in energy and momentum but also into spin and orbital degrees of freedom. A wide range of materials have been investigated for clarifying their electronic dynamics, such as the recombination of the superconducting quasiparticles[21,22], fluctuating charge density waves[23], collapse of long-range order[24,25], and coupling with optical phonons[25–27]. These results, which are inaccessible from equilibrium state, contributed to the deeper understanding of exotic quantum states, especially those with short lifetimes.

Here, we use TARPES to track the ultrafast dynamics of electronic nematicity in FeSe. By combining detwinned crystals with a linear-polarized probe laser, we can selectively obtain the electrons of $xz$ and $yz$ orbitals (Supplementary Note 3). With this TARPES setup[28] (Fig. 1e), the ultrafast dynamics of the nematic Fermi surface and the orbital-dependent carrier dynamics can be visualized.

## Results

**Ultrafast dynamics of nematic Fermi surface.** Immediately after photo-excitation ($t = 120$ fs), the hole bands around the $\Gamma$ point along $k_x$ and $k_y$ exhibit remarkable momentum shifts with the opposite signs, and take comparable $k_F$ values as indicated by the red and blue arrows in Fig. 2a, b. This observation suggests that the elliptical Fermi surface quickly changes to circular by the photo-excitation, thus indicating the melting of the nematic order. Fig. 2c displays the fluence ($F$)-dependence, where the shift of $k_{Fy}$ ($\Delta k_{Fy}$) at $t = 120$ fs gradually increases as a function of $F$ (weak-excitation regime), and saturates at $F > F_c = {\sim}200$ μJ cm$^{-2}$ where the isotropic Fermi surface is attained (strong-excitation regime).

Here we track the time dependence of $\Delta k_{Fy}(t)$ for respective $F$ (Fig. 2d). As shown in Fig. 2c, $\Delta k_{Fy}(t)$ indicates the sudden decrease at $t \approx 120$ fs representing the melting of nematicity, followed by the subsequent recovery in ~1 ps. The overall picture of this transient Fermi surface for $F > F_c$ is shown in Fig. 2e. We further find that the recovery clearly becomes faster for $F > F_c$, and some modulated feature appears. The time dependences of $k_{Fx}$ and $k_{Fy}$ for 220 μJ cm$^{-2}$, where the modulation appears most strongly, are presented in Fig. 2f. These data indicate that the $C_2$ anisotropy in $k_F$ is completely suppressed ($k_{Fx} \approx k_{Fy}$) within the time resolution (250 fs), followed by an anomalous hump in the recovery. These behaviors of $k_{Fx}$ and $k_{Fy}$ can be reproduced by the functions including the damped oscillation term in the form of $k_F(t) = k_{F0} + k_{F1}\exp(-t/\tau_1) + k_{F2}\exp(-t/\tau_2) + k_{F3}\exp(-t/\tau_3)\cos(2\pi t/t_p)$ convoluted by the Gaussian of the time resolution, with common values of $\tau_1 = 830 \pm 50$ fs, $\tau_2 > 80$ fs, $\tau_3 = 550 \pm 50$ fs, and $t_p = 1.4 \pm 0.05$ ps. The observed anti-phase oscillation of $k_{Fx}$ and $k_{Fy}$ directly represents the Pomeranchuk-type oscillation of Fermi surface[29], being intensively discussed as the fundamental excitation in the electronic nematic state. The time scale of the oscillatory response (1.4 ps, 3.1 meV) is considerably slow as compared to the coherent $A_{1g}$ optical phonon (190 fs, 22 meV),

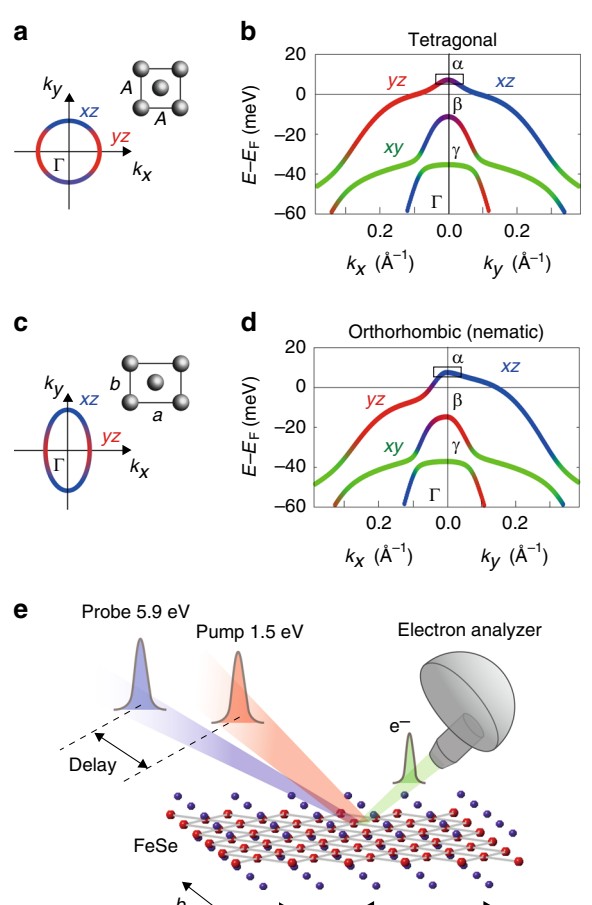

**Fig. 1** Electronic structure of FeSe and experimental setup for TARPES. **a** Schematic Fe lattice and Fermi surface around the $\Gamma$ point in the tetragonal phase. $x$ and $y$ are coordinates along the crystal axes of the orthorhombic setting $a$ and $b$ ($a > b$), respectively. **b** Band dispersions and orbital characters in the tetragonal phase obtained by a band calculation including the spin–orbit coupling[17]. $\alpha$, $\beta$, and $\gamma$ denote the outer, middle, and inner hole band, respectively. **c, d**, The same as **a** and **b** but for the orthorhombic phase. For reproducing the ARPES results (Supplementary Note 1), the spin–orbit coupling and orbital polarization were included in the band calculations in **d** (ref. [17]). **e** Schematic experimental geometry of TARPES on detwinned bulk FeSe

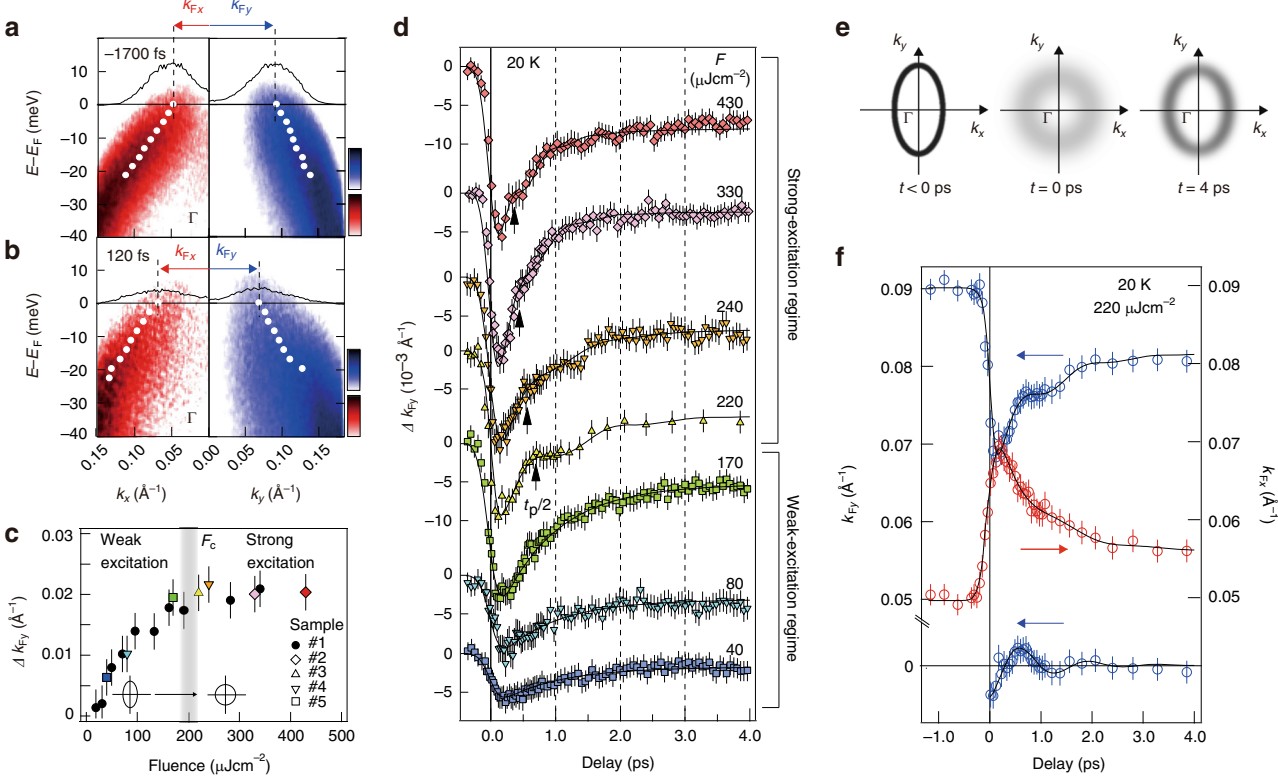

**Fig. 2** Ultrafast transformation of the nematic Fermi surface. **a** $E$–$k$ images in a logarithmic color scale obtained by $p$-polarized probe laser of $h\nu = 5.9$ eV along $k_x$ (left panel) and $k_y$ (right panel) axes at 20 K, before photo-excitation of $F = 220$ μJ cm$^{-2}$ ($t = -1700$ fs). Black curves represent the momentum-distribution curves (MDCs) at the $E_F$ and the broken black lines indicate their peak positions. White markers show the band dispersions obtained from the MDC peaks. **b** The same as **a** but after photo-excitation ($t = 120$ fs). **c** $F$ dependence of the $k_F$ shift along $k_y$ ($\Delta k_{Fy}$) at $t = 120$ fs. $F_c$ represents the $F$ where the $C_4$ symmetric Fermi surface is attained after the photo-excitation. **d** $F$ dependence of $\Delta k_{Fy}(t)$. The data set was obtained at 20 K from five single crystals as indicated by the different markers in **c**. **e** Schematics of the Fermi surface around the $\Gamma$ point for $t < 0$ ps, $t = 0$ ps, and $t = 4$ ps deduced from the transient $k_{Fx}$ and $k_{Fy}$, and the width of the MDCs at the $E_F$. **f** Transient $k_{Fy}$ (blue open circles) and $k_{Fx}$ (red open circles) as a function of delay time with the fitting functions (black curves). Damped oscillation in $k_{Fy}$ was extracted by subtracting the exponential decay components, as shown in the bottom

which is known to strongly couple to the electronic state in this system[25–27]. Their possible interplay is unfortunately hidden in the present TARPES data, possibly due to the duration of the pump pulse (170 fs) comparable to the time period of A$_{1g}$ mode (190 fs) that tends to vanish the coherent oscillation.

**Orbital-dependent carrier dynamics.** Based on the behavior of transient Fermi surface, we focus on the orbital-dependent carrier dynamics. With $p$-polarized probe pulse, we obtain the energy-distribution curves (EDCs) for $xz$ and $yz$ electrons around the $\Gamma$ point by integrating $k_y$ and $k_x$ in $0.00 \pm 0.04$ Å$^{-1}$, respectively (see Supplementary Note 3 for experimental settings). The main peak of EDC around $-18$ meV in Fig. 3a, b, e, f corresponds to the top of the middle ($\beta$) hole band sinking below $E_F$, predominantly of $yz$ orbital character (Fig. 1d). In the weak-excitation regime of $F = 40$ μJ cm$^{-2}$ (Fig. 3a, b), the main peak gets rapidly suppressed, and electrons are excited toward the unoccupied state. Here we track the evolution of the corresponding photo-electron intensities $\Delta I(t)$ at $E - E_F = 7.5 \pm 2.5$ meV and $k = 0.00 \pm 0.04$ Å$^{-1}$, i.e. black rectangles in Fig. 1b, d. Figure 3c, d shows that $\Delta I(t)$ for $xz$ and $yz$ exhibit the similar exponential decay function with two time constants 850 fs and >80 ps, thus indicating the mostly equivalent relaxation processes of both orbitals. However, we note that the excited tail intensity of EDCs at $E > E_F$ is very low for $yz$, being consistent with the predominantly $xz$ character of the outer ($\alpha$) hole band top at $\Gamma$ (Fig. 1d). Substantial spectral weight depletion in the $yz$ states after pumping might be attributed to the

photoexcited $yz$ electrons which are partly trapped at the M point. Because of the semi-metallic electronic structure, some part of electrons excited by 1.5 eV photons at $\Gamma$ may quickly gather around the electron bands at M. The momentum-dependent sign-inversion of orbital polarization[17] realizes the $yz$ dominated electron pocket near $E_F$ at M, which may work as the reservoir for photoexcited $yz$ electrons. To fully understand these dynamics, the TARPES covering the whole Brillouin zone is desired.

In the strong-excitation regime, the photo-response changes drastically. The EDCs in Fig. 3e, f show that the photoexcited states at $E > E_F$ also appear in $yz$, indicating that the $C_4$ isotropic state (Fig. 1b) is achieved by the strong photo-excitation ($F = 220$ μJ cm$^{-2}$). On the other hand, the excited intensity of $xz$ shows a nonmonotonic relaxation which keeps increasing from $t = 120$ to 700 fs as indicated by the black arrow in Fig. 3e, being markedly different from $yz$. As shown in Fig. 3g, h, $\Delta I(t)$ of $xz$ exhibits the retarded maximum at $t_{ret} = \sim 700$ fs, whereas the $yz$ electrons show the exponential decay more or less similar to the weak-excitation regime, with the initial maximum at $\sim 250$ fs. These contrastive behaviors solely depend on the orbital characters, not on experimental configuration (Supplementary Notes 4 and 5). To discuss the retardation behavior, the $F$ dependence of $\Delta I(t)$ for $xz$ is shown in Fig. 3i. In the weak-excitation regime ($F < F_c$), $\Delta I(t)$ curves commonly show the simple relaxation with the maximum around 250 fs. With increasing $F$, the retardation suddenly shows up at $F \approx F_c$. Its time scale estimated by $t_{ret}$ is 700 fs at $F \approx F_c$ and gradually decreases to 350 fs by increasing $F$ to 430 μJ cm$^{-2}$.

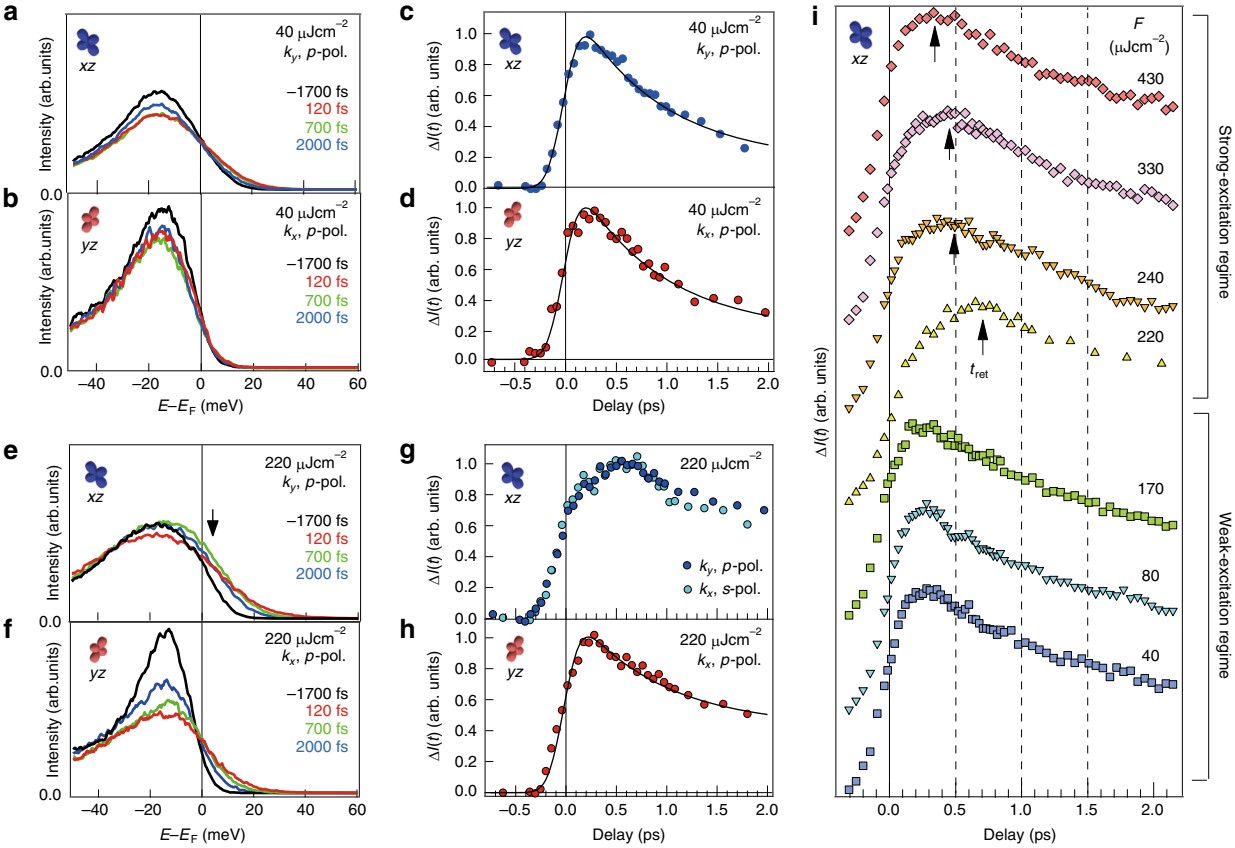

**Fig. 3** Orbital-dependent dynamics of the photoexcited electrons at the $\Gamma$ point. **a** EDCs around the $\Gamma$ point ($k = 0.0 \pm 0.04 \text{ Å}^{-1}$) obtained from the $E$–$k$ images at 20 K along $k_y$ at $t = -1700$, 120, 700, and 2000 fs, in the $p$-polarized setting. The fluence of the photo-excitation was $F = 40 \text{ μJ cm}^{-2}$. **b** The same as **a** but obtained along $k_x$ axis. $xz$ ($yz$) electrons are dominantly observed in **a** (**b**). **c** The photo-electron intensities normalized by that before photo-excitation, $\Delta I(t)$, at the $\alpha$ band top ($E$–$E_F = 7.5 \pm 2.5 \text{ meV}$) around the $\Gamma$ point ($k_y = 0.0 \pm 0.04 \text{ Å}^{-1}$), obtained by the $p$-polarized probe laser and the pump fluence of $F = 40 \text{ μJ cm}^{-2}$. The black curve represents the fitting function assuming the double exponential components with the time constants of 850 fs and 80 ps. **d** The same as **c** but at $k_x = 0.0 \pm 0.04 \text{ Å}^{-1}$. **e–h** The same as **a–d** but with the pump fluence of $F = 220 \text{ μJ cm}^{-2}$. The light blue markers in **g** represent the $\Delta I(t)$ of $xz$ obtained by an $s$-polarized probe laser (Supplementary Note 4). **i** $F$ dependence of the normalized $\Delta I(t)$ for $xz$. The black arrows represent the time of the retarded maxima $t_{ret}$ for $F > F_c$

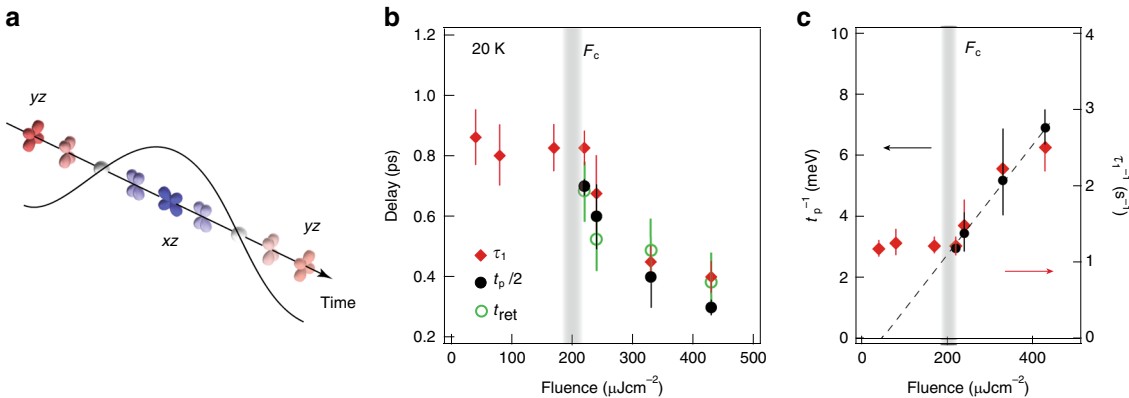

**Fig. 4** Pump-fluence dependence of the time scales for the orbital and nematicity dynamics. **a** Illustration of the orbital excitation in FeSe. **b** $F$ dependences of $t_{ret}$ (open green circles), $t_p/2$ (filled black circles) and $\tau_1$ (filled red diamonds). **c** $t_p^{-1}$ (filled black circles) and $\tau_1^{-1}$ (filled red diamonds) plotted as functions of $F$. The dotted line is a linear function derived from the data at $F > F_c$

**Fluence-dependent dynamics of electronic nematicity**. Here we summarize the dynamics of the electronic nematicity in Fig. 4b. By fitting $\Delta k_{Fy}(t)$ curves in Fig. 2d (Supplementary Note 6), we plot $\tau_1$ and $t_p/2$ for respective $F$. $\tau_1$ is the exponential decay time indicating the quick recovery of nematicity. It shows a constant value (~800 fs) in the weak-excitation regime and a rapid

decrease above $F > F_c$. The time scale of the Fermi surface oscillation indicated by $t_p/2$, which only appears in $F > F_c$, also rapidly decreases from 700 to 300 fs as increasing $F$. It shows that the oscillation gets more severely damped and hard to observe at high $F$. We also overlay the time scale of the retarded maximum in $xz$ component $t_{ret}$. As a result, $\tau_1$, $t_p/2$, and $t_{ret}$

similarly show the maximum values at $F_c = 220\,\mu J\,cm^{-2}$ that monotonically decrease with increasing $F$, while keeping the common relation $\tau_1 \approx t_{ret} \approx t_p/2$. The relation $t_{ret} \approx t_p/2$ implies that the orbital-dependent carrier dynamics is synchronized with the short-life nematic Fermi surface oscillation. We note that the transient Fermi surface at $t_p/2$ ($\approx t_{ret}$) is more elliptical than that expected without the oscillatory response. Such an overshoot of the nematicity in Fermi surface should also appear in the orbital-dependent carrier dynamics. In the process relaxing back from $C_4$ isotropic to $C_2$ nematic ground state, the electrons at the band top (black rectangle in Fig. 1b, d) change their orbital characters from "(nearly) $xz/yz$ degenerate" to "predominantly $xz$". The retarded maximum in $I(t)$ for $xz$ can be thus regarded as an indication of the orbital redistribution from $yz$ to $xz$ (Fig. 4a). The synchronized responses in the Pomenranchuk Fermi surface oscillation and orbital-dependent carrier dynamics thus represent the nematic-orbital excitation.

## Discussion

Now we discuss the nematic-orbital excitation obtained in the present TARPES results by comparing with the nematic dynamics in thermal equilibrium as probed by recent Raman scattering measurements[15,30]. The electronic Raman spectra of $XY$ symmetry ($X$ and $Y$ are coordinates along the crystal axes of the tetragonal setting) show the characteristic quasi-elastic peak (QEP) evolving toward $T_s$ on cooling the temperature ($T$), discussed in terms of nematic susceptibility enhancement[15,30]. The QEP rapidly diminishes at $T < T_s$, on the other hand, and a gap opens in the $XY$ Raman spectra, thus indicating the suppression of low-energy nematic excitations[30]. These behaviors are reminiscent of the nematic-orbital excitation observed by TARPES, where the peculiar slowing down behavior shows up in $F > F_c$, and the excitation itself suddenly disappears in $F < F_c$. The nematic fluctuation is incoherent in nature, however, by instantaneously triggering the dissolution of the nematic state, it may be appearing as the heavily damped oscillatory response in the time domain.

Further insight of the peculiar $F$ dependence can be obtained by plotting $t_p^{-1}$ and $\tau_1^{-1}$ (Fig. 4c). These values show a more or less $F$-linear behavior at $F > F_c$, indicating the critical slowing down. At $F = F_c$, $t_p^{-1}$ decreases down to 3.1 meV. In $F < F_c$, as already mentioned, the $k_F$ oscillation as well as the anomaly in the $xz$ orbital response disappear, and $\tau_1^{-1}$ becomes constant. In the $XY$ Raman spectrum, the $T$-linear behavior was found in the inverse of the QEP intensity above $T_s$ (ref. [15]), indicative of the Gaussian fluctuation evolving in this regime. Similarly, the elastoresistivity measurement had also revealed the existence of electronic nematic fluctuation at $T > T_s$ interpreted as the Curie–Weiss-like nematic susceptibility[30]. Through the analysis of the $T$-dependent nematic susceptibility in the form of $|T - T_0|^{-1}$, the authors discuss the mean-field transition temperature $T_0$ in terms of the ideal nematic transition purely driven by electrons without any influence of lattice[15,30]. For FeSe, $T_0$ was estimated to be far below $T_s$, i.e. 8, 20 (ref. [15]), and 30 K (ref. [31]). The Curie–Weiss-like behavior of $t_p^{-1}$ and $\tau_1^{-1}$ toward $F \approx 40 \pm 20\,\mu J\,cm^{-2}$, i.e. much smaller than $F_c = 220\,\mu J\,cm^{-2}$, may be reflecting that the base temperature of the TARPES measurements (20 K) is close to $T_0$. This scenario is also consistent with the initial photo-response of $\Delta k_{Fy}$ with small threshold ($<30\,\mu J\,cm^{-2}$, see Fig. 2c). These results indicate that the electronic nematiciy in the initial ultrafast regime (~120 fs) shows the flexible photo-reaction by decoupling from the lattice. Our analysis on the transient electronic temperature ($T_e$) (Supplementary Note 7) indeed shows that $T_e$ immediately reaches $88 \pm 2$ K at 120 fs and then decreases in less than 1 ps (Supplementary

Fig. 6). For $t > 3$ ps, it becomes nearly constant at ~45 K, indicating the realization of quasi-equilibrium state where the temperatures of electrons and lattice become equivalent through the electron–lattice coupling[32]. The maximum lattice temperature is thus much lower than $T_s$ ($= 90$ K), showing that the lattice stays orthorhombic. We also note that the reduction of the lattice orthorhombicity is known to occur in a much slower time scale (e.g. ~30 ps) with a much higher pump fluence (e.g. 3.3 mJ cm$^{-2}$) for BaFe$_2$As$_2$ (ref. [33]).

The present results show that the femtosecond photon pulse can perturb the electronic nematic order and instantly decouple it from the lattice. Only in the strong-excitation regime where the nematic state is completely destroyed, there appears the peculiar dynamical process involving the orbital redistribution and short-life Pomenranchuk-type Fermi surface oscillation. This behavior is seemingly related to the critical nematic fluctuation; nevertheless, future theoretical studies on non-equilibrium critical phenomena are highly necessary. The recovery time scale of the nematic Fermi surface is strongly correlated with the short-life $k_F$ oscillation ($\tau_1 \approx t_p/2$), which also awaits investigations on the dynamics of fluctuation and dissipation in non-equilibrium states. Experimentally, further studies on the nematic dynamics around the quantum critical point in the FeSe$_{1-x}$S$_x$ system[31] and the coherent nematic resonance mode in the superconducting state is highly desired. Systematic time-resolved diffraction measurements will also help understanding the possible interplay among the nematic excitation and the optical/acoustic phonons[26,34]. Ultrafast photo-excitation adds to the possibilities for understanding and manipulating the large-amplitude electronic fluctuations associated with phenomena such as unconventional superconductivity, exotic magnetism, thermopower enhancement, and so on.

## Methods

**Sample preparations.** High-quality single crystals of FeSe were grown by the vapor transport method. A mixture of Fe and Se powders was sealed in an evacuated SiO$_2$ ampoule together with KCl and AlCl$_3$ powders[16]. The transition temperatures of the single crystals were estimated to be $T_s = 90$ K and $T_c = 9$ K from the electrical resistivity measurements. We showed the data obtained from five single crystals of FeSe which were synthesized less than 2 months before the TARPES measurements.

**Time and angle-resolved photoemission measurements.** The TARPES measurements were done at ISSP, the University of Tokyo[28]. The laser pulse (1.5 eV and 170 fs duration) delivered from a Ti:Sapphire laser system operating at 250 kHz repetition (Coherent RegA 9000) was split into two branches: one is used as a pump and the other was up-converted into 5.9 eV and used as a probe to generate photoelectrons. The delay origin $t = 0$ ps and time resolution (250 fs) were determined from the pump-probe photoemission signal of graphite attached near the sample. The photoelectrons were collected by a VG Scienta R4000 electron analyzer. The $E_F$ and the energy resolution (20 meV) were determined by recording the Fermi cutoff of gold in electrical contact to the sample. To detwin the single crystals, we applied an in-plane uniaxial tensile strain[17,19], which brings the orthorhombic $a$ axis ($a > b$) along its direction below $T_s$. We chose $s$ and $p$ polarizations to separately observe the $xz$ and $yz$ orbital electrons (see Supplementary Note 3 for the details of experimental geometry and selection rule). Samples were cleaved in situ at room temperature in an ultrahigh vacuum of $5 \times 10^{-11}$ Torr.

## Data availability
The datasets generated during and/or analyzed during the current study are available from the corresponding author on reasonable request.

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

## Acknowledgements

We thank M. Imada, Y. Yamaji, Y. Gallais and I. Paul for valuable discussions. We acknowledge H. Kontani and Y. Yamakawa for valuable discussions and band calculations. This research was supported by the Photon Frontier Network Program of the MEXT, the CREST project of the JST (Grant Number JPMJCR16F2), Grant-in-Aid for Scientific Research (KAKENHI) (Grant Nos. 15H03687, 15H02106, 15KK0160, 18H01177, 18H05227 and 18H01148) and on Innovative Areas "Topological Material Science" (No. 15H05852) from Japan Society for the Promotion of Science (JSPS).

## Author contributions

T. Shimojima and K.I. designed the research. T. Shimojima, Y.S., A.N., N.M. and Y.I. performed the TARPES measurements and analyzed the data. S.K., T. Shibauchi, and Y.M. synthesized the single crystals. Y.I. and S.S. set up the TARPES apparatus. T. Shimojima wrote the paper with inputs from S.K., T. Shibauchi, Y.M., Y.I., S.S. and K.I.

## Additional information

**Competing interests:** The authors declare no competing interests.

