## [Peer Review File · Nature Communications]

Reviewers' comments:

Reviewer #1 (Remarks to the Author):

Manuscript "Ultrafast nematic-orbital excitation in FeSe" reports orbitally resolved time-dependent ARPES measurements in FeSe after photo-excitation with a 1.5 eV femtosecond laser pulse. ARPES allows to directly extract Pomeranchuk-type Fermi surface (FS) deformation (from circular to elliptical) arising from nematic order. Varying the probe light polarization allows to (approximately) measure the orbitally resolved occupation numbers. Since orbital nature changes from tetragonal to orthorhombic state, authors can make conclusion about nematicity from this observation as well.

Main results are the rapid melting of nematic order (or more precisely the rapid vanishing of the FS deformation) on 120 fs timescale due to the optical pump. This is followed by a recovery on ps timescales, that interestingly exhibits an oscillatory behavior for pumps above a certain threshold (of $F_c = 200 \mu\text{J}/\text{cm}^2$). Second main result is the different behavior of photo-excited electrons with xz and yz orbital character. Specifically, at (and above) threshold F_c the xz orbital occupation continues to increase (for ≤ 500 fs) beyond the initial laser induced increase, which is in contrast to the occupation of the yz orbitals. An analysis of timescales $\tau_{1,2}$, t_{osc} extracted from experiment, let's the authors conclude that the electronic dynamics in the sub-ps regime is decoupled from the lattice.

The presented results are interesting and on a timely subject of nematic order and its origins in FeSe. However, while the data is clearly presented, the analysis and in particular the conclusions that the authors draw from their data remains mostly speculative and semi-quantitative. While the experimental data may contain information that may lead to new insights about FeSe, the present manuscript does not sufficiently flesh out and justify any novel insights. I suggest that the authors point out more clearly what are the novel insights into FeSe that follow from their non-equilibrium study. Therefore, the paper is not of sufficient interest for the general reader (beyond the ultrafast community) and I cannot recommend publication of the manuscript in its present form at Nature Communications.

1) For example, the author's analysis on the origin of the oscillatory behavior in Δk_F remains rather inconclusive. The authors write what they think is not the origin (optical phonons, order parameter oscillation), but not what they think the origin is.

2) Another example, is the origin of the different behavior of the orbital occupations, and in particular the discontinuous jump of t_{ret} at threshold. The main explanation (orbital flipping) that the authors offer to the reader, which is that orbital occupations oscillate between xz and yz, is not justified in the manuscript. In particular, based on the authors explanations, one would expect some effect of the orbital flipping to be visible in the yz occupations as well. Could authors please describe and clarify in more detail the process they call 'orbital flipping' and how they conclude that it occurs?

3) Identifying t_p^{-1} with the equilibrium quantity Γ from Raman scattering in Ref.[15] simply because both timescales are approximately $(3 \text{ meV})^{-1}$ is not justified either. Since one of the main conclusions of the paper, which is that the electronic nematic dynamics is uncoupled from the lattice (which seems reasonable), is based on this identification, this must be underpinned with theory/analysis more thoroughly.

4) Another minor question is related to the meaning of k_F in non-equilibrium. The electronic distribution function is clearly broadened and may not even be thermal. Looking at Fig.1(b), it looks as if there should be a substantial (systematic) error bar being associated with extracting k_F in non-equilibrium, but the data in panel (d) contain no error bars. It would be useful if authors try to estimate the hypothetical final temperature of the electronic system based on the energy they

deposit into the system, and compare this with T_S .

Reviewer #2 (Remarks to the Author):

The authors present a time-resolved ARPES study of detwinned FeSe, and report on the topic of pump-probe effects of electronic nematicity in this material. This is potentially an interesting study on the out-of-equilibrium effects of electronic nematicity, which would be novel to the field. However, there are several issues that prevent me from recommending publication in its current form.

1. Could the authors comment on the effect of transient heating? Since the pump pulse necessarily deposits energy unto the sample, which very likely heats the sample up, it is hard to disentangle whether the observed disappearance of nematicity is due to intrinsic excitation out of this electronic phase at a temperature below T_S or simply the effect of having raised the electronic temperature to above T_S in this material. The transient heating would also have a fluence dependence where beyond a critical fluence the sample would always be pumped above T_S , and a saturation of the change in k_f is observed. This is an important issue regarding the argument that the pumped response probes the electronic nematicity decoupled from the lattice degree of freedom because in the case of transient heating the disappearance of electronic nematicity is observed on a tetragonal lattice via raised temperature, and in the case without transient heating the observed disappearance of electronic nematicity is observed on an orthorhombic lattice at base temperature.

2. Going along with the last point, one way to disentangle this is perhaps to look at the EDCs after pumping, such as in Fig. 3 and the supplement figures. Could the EDCs be fitted to a functional form with a Fermi-Dirac function convolved with the instrumental resolution to extract the electronic temperature as a function of delay time? If the change is due purely to electronic dynamics, one may expect the EDCs to show a peak or at least hump-like feature above EF where the authors argue that the xz states lie, and where the pump excites the states below EF to. The EDCs in the figure currently look perhaps more like thermally broadened EDCs with a single peak below EF. Could some fittings under the two scenarios resolve this issue?

3. Along the same lines, comparing Fig. 3a and 3b, if the pumping excites xz electrons to unoccupied xz states above EF, where do the pumped yz electrons go in panel b? It seems that there is substantial spectral weight depletion after pumping in the yz states. Then where does it go if it cannot go into the unoccupied xz states?

4. There is an oscillatory component in the delay response for higher fluences. This, if real, would not be due to transient heating. Could the authors discuss more the potential origin of this mode?

5. In ref. 25, where a similar tr-ARPES is done on FeSe films in the same fluence regime, a clear coherent oscillation is reported. But such oscillation is not seen in the current set of data. Could the authors clarify in the text the reason of this inconsistency? It is likely due to the different regimes of time/energy resolution used, but would be good for the general public to know.

6. Around the Gamma point, as has been reported, there is substantial spin-orbit coupling that mixes the xz and yz orbitals. How does this energy scale affect the dynamics of the pump-probe process (ie transitions between xz/yz states)?

Reviewer #3 (Remarks to the Author):

The authors of the article performed time resolved ARPES experiments on FeSe de-twinned by external stress. They monitored the dynamics of electronic states in the nematic phase along the two high symmetry axis of the crystal. The data indicate that photoexcitation with moderate fluence can disrupt the nematic order on an ultrafast timescale. The melting and recovery time are comparable to the period of a coherent modulation appearing at intermediate fluence. This article is very interesting and will have high impact in the scientific community. It is clearly written and self-consistent. The large amount of experimental data that are discussed will provide an important reference for future works.

My only criticism regards the speculative attempt to identify the fluence dependence of the timescales with a critical behavior. Could the author explain what is a quantum critical point? A hypothetical nematic phase that, if decoupled from the lattice, would display a transition temperature of 20 K does not lead to a quantum critical point. On which basis such hypothetical phase would lead to fluctuations affecting linearly the electronic timescale if the fluence overcomes the threshold value?

Beside the necessary clarification on this critical vs non-critical behavior, I recommend the publication of the present article in Nature Communication.

Reply to the Reviewer #1:

The presented results are interesting and on a timely subject of nematic order and its origins in FeSe. However, while the data is clearly presented, the analysis and in particular the conclusions that the authors draw from their data remains mostly speculative and semi-quantitative. While the experimental data may contain information that may lead to new insights about FeSe, the present manuscript does not sufficiently flesh out and justify any novel insights. I suggest that the authors point out more clearly what are the novel insights into FeSe that follow from their non-equilibrium study. Therefore, the paper is not of sufficient interest for the general reader (beyond the ultrafast community) and I cannot recommend publication of the manuscript in its present form at Nature Communications.

We sincerely thank Reviewer 1 for carefully reading our manuscript and giving us important suggestions and comments. By answering to all his/her questions and comments as described below, we believe that the conclusion of our work and the derived novel insights are now much more clearly presented.

1) For example, the author's analysis on the origin of the oscillatory behavior in Δk_{Fy} remains rather inconclusive. The authors write what they think is not the origin (optical phonons, order parameter oscillation), but not what they think the origin is.

We thank the Reviewer for his/her important comment. As the Reviewer pointed out, the discussion on the oscillatory behavior in Δk_{Fy} was rather inconclusive. The most important aspect is that the observed anti-phase oscillation of Δk_{Fx} and Δk_{Fy} , as shown in Fig. 2f, is the direct representation of Pomeranchuk-type Fermi surface oscillation. To make this point clear, we added the description below in line 8, page 4.

The observed anti-phase oscillation of k_{Fx} and k_{Fy} rather directly represents the Pomeranchuk-type oscillation of FS²⁹, being intensively discussed as the fundamental excitation in the electronic nematic state.

Accordingly, we added new reference number 29 as following.

[29] Pomeranchuk, I. Ia. On the stability of a Fermi liquid. *J. Exp. Theor. Phys.* **8**, 361 (1959)

In addition, we modify the discussion part. As described in the previous manuscript, electronic Raman scattering measurements [PNAS **113**, 9177 (2016)] reported the critical behavior in the nematic susceptibility with the XY-symmetry for FeSe. By further referring to another recent study [arXiv:1710.09892], we can more deeply discuss the similarities between the nematic dynamics obtained

by the electronic Raman scattering and our TARPES. In the Raman scattering study, the critical enhancement of nematic susceptibility and corresponding quasi-elastic peak (QEP) in the XY spectrum are observed in the tetragonal phase, on cooling toward the structural transition temperature (T_s). It implies the presence of the critical nematic fluctuation in $T > T_s$. On cooling below $T < T_s$, on the other hand, the QEP Raman intensity rapidly diminishes, and a gap opens in the XY spectrum indicating the sudden suppression of low-energy excitations [Fig. 1(b) in arXiv:1710.09892]. On further cooling below the superconducting transition temperature (T_c), they find a peak appearing at 3.6 meV, which they assign to the nematic resonance mode acquiring the coherence in the superconducting state. First we note that the energy of this nematic mode (3.6 meV) is fairly close to that of the Pomeranchuk-type FS oscillation (3.1 meV for 220 $\mu\text{J}/\text{cm}^2$) obtained by TARPES, thus suggesting the similarity in the origin. Nevertheless, we have to also note that our measurements are done in the non-superconducting state ($T \sim 20 \text{ K} > T_c$), where the coherent nematic mode does not exist. Our present interpretation is that the observed short-life FS oscillation should be associated with the QEP (i.e. nematic fluctuation) in the Raman study, since the T -dependence of QEP is very similar to the F -dependence of Δk_{FY} oscillation; apparent only in $F > F_c$, and suddenly disappears in $F < F_c$. The nematic fluctuation should be of course incoherent in nature, however, we consider that by instantaneously triggering the dissolution of the nematic state, it can appear as the heavily-damped oscillatory response in the non-equilibrium time domain.

With these facts, we believe that we can offer the specific microscopic picture of the nematic excitation / fluctuation (i.e. the Pomeranchuk-type FS oscillation accompanying the orbital redistribution), which has been long discussed in the community due to its ubiquitous nature, but without being well identified.

To account for the Reviewer's advices, we modified the sentences in line 4, page 6 of the revised manuscript.

The nematic-orbital excitation obtained in the present TARPES shows a striking resemblance with the nematic dynamics in thermal equilibrium as probed by the recent Raman scattering measurements.^{15,30} The electronic Raman spectra of XY symmetry (X and Y are coordinates along the crystal axes of the tetragonal setting) show the characteristic quasi-elastic peak (QEP) evolving toward T_s on cooling the temperature (T), discussed in terms of nematic susceptibility enhancement.^{15,30} The QEP rapidly diminishes at $T < T_s$, on the other hand, and a gap opens in the XY Raman spectra thus indicating the suppression of low-energy nematic excitations (Ref. 30). These behaviors are reminiscent of the nematic-orbital excitation observed by TARPES, where the peculiar slowing behavior shows up in $F > F_c$, and the excitation itself suddenly disappears in $F < F_c$. The XY Raman spectrum further reveals a peak at 3.6 meV in the superconducting state ($T < T_c$), which is interpreted in Ref [30] as the nematic resonance mode acquiring the coherence by the superconducting gap opening. This energy scale is fairly close to that of the damped k_{F} oscillation (3.1 meV) observed by TARPES near F_c , thus suggesting the similarity in its origin. With

these facts, we presently consider that the nematic-orbital excitation obtained by TARPES should be associated with the QEP (i.e. nematic fluctuation) in the Raman study. The nematic fluctuation is incoherent in nature, however, by instantaneously triggering the dissolution of the nematic state, it may be appearing as the heavily-damped oscillatory response in the time domain.

We also added the reference number 30 as follows.

[30] Zhang, W.-L. *et al.*, Stripe quadrupole order in the nematic phase of FeSe_{1-x}S_x. arXiv:1710.09892.

2) Another example, is the origin of the different behavior of the orbital occupations, and in particular the discontinuous jump of t_{ret} at threshold. The main explanation (orbital flipping) that the authors offer to the reader, which is that orbital occupations oscillate between xz and yz , is not justified in the manuscript. In particular, based on the authors explanations, one would expect some effect of the orbital flipping to be visible in the yz occupations as well. Could authors please describe and clarify in more detail the process they call 'orbital flipping' and how they conclude that it occurs?

We thank the Reviewer's valuable suggestion. We agree that our interpretation of orbital-dependent carrier dynamics regarding t_{ret} should be more carefully and precisely explained. Here we would like to emphasize that t_{ret} is nearly identical to $t_p/2$ (i.e. half of k_F oscillation period) for all pump fluences ($F > F_c$). For the k_F oscillation, we note that the transient FS at $t_p/2$ is more elliptical than that expected without the oscillatory response. Such an overshoot of the nematicity should also appear in the orbital dependence of the carrier dynamics. In the process relaxing back from C_4 isotropic to C_2 nematic ground state, the electrons at the band top (black rectangles in Fig. 1 b,d) change their orbital characters from "degenerate xz/yz " to "predominantly xz ". According to this consideration, we interpret that the retarded maximum in $I(t)$ for xz should correspond to the orbital redistribution from yz to xz orbital (Fig. 4a), accompanied by the overshoot of the FS nematicity at $t_p/2$.

In this scenario, just as the Reviewer pointed out, there should be the counterpart decrease in the yz intensity at around t_{ret} . However, we could not observe clear evidence for the anomalous dynamics in the yz occupation. Since we are looking at the unoccupied state (7.5 ± 2.5 meV above E_F), there is always the decaying component of photoexcited electrons relaxing toward $E < E_F$, appearing as the rapid decrease of intensity with the time constant of ~ 850 fs (fairly close to t_{ret} at $F = 220 \mu\text{Jcm}^{-2}$). The counterpart decrease expected in $I(t)$ for yz may be overlapped by this process, thus making it difficult to be separately observed.

Through the above discussion, we have come to modify the term 'orbital flipping' to 'orbital redistribution', which should be more precisely representing the phenomena involving the change in the relative xz and yz occupations.

We added the sentences below in line 21 page 5 of the revised manuscript.

We note that the transient FS at $t_p/2$ ($\approx t_{\text{ret}}$) is more elliptical than that expected without the oscillatory response. Such an overshoot of the nematicity in FS should also appear in the orbital-dependent carrier dynamics. In the process relaxing back from C_4 isotropic to C_2 nematic ground state, the electrons at the band top (black rectangle in Fig. 1 b,d) change their orbital characters from “(nearly) xz/yz degenerate” to “predominantly xz ”. The retarded maximum in $I(t)$ for xz can be thus regarded as an indication of the *orbital redistribution* from yz to xz (Fig. 4a). The synchronized responses in the FS oscillation and orbital-dependent carrier dynamics thus represent the *nematic-orbital excitation*.

According to this modification, we revised the words “orbital flipping” into “orbital redistribution” in line 21, page 7 of the revised manuscript.

3) Identifying t_p^{-1} with the equilibrium quantity Γ from Raman scattering in Ref.[15] simply because both timescales are approximately $(3 \text{ meV})^{-1}$ is not justified either. Since one of the main conclusions of the paper, which is that the electronic nematic dynamics is uncoupled from the lattice (which seems reasonable), is based on this identification, this must be underpinned with theory/analysis more thoroughly.

We thank the Reviewer for the valuable suggestion. We agree that there is no strong justification on directly comparing Γ of QEP in static Raman studies with the present t_p^{-1} . From the result we believe there is some unknown relationship between them, however, at present we lack any theoretical support. For this reason, we modified the part describing the similarity in the behaviors of nematic dynamics obtained by Raman studies and TARPES, as mentioned in Reply 1). Regarding the timescale, we now compare t_p^{-1} (3.1 meV) with the peak energy of the nematic resonance Raman mode (3.6 meV) appearing in the superconducting state (arXiv:1710.09892), to discuss its possible similarity in the origin. Regarding the critical (F -linear) behavior, we simply focus on the T -linear behavior of the inverse of the QEP Raman intensity (interpreted as the nematic susceptibility), suggesting the critical enhancement of the nematic fluctuation toward the electronic nematic transition at $T_0 \sim 20$ K (much below $T_s = 90$ K) [PNAS **113**, 9177 (2016)]. The critical behavior of t_p^{-1} toward $F \approx 40 \mu\text{Jcm}^{-2}$, i.e. much smaller than F_c , may be reflecting that the base temperature of the TARPES measurements (20 K) is close to T_0 . This similarity suggests that the dynamics of the nematicity in TARPES is seemingly electronic and thus decoupled from lattice.

Regarding the decoupling of the electronic system from the lattice, we also added some discussion based on the transient electronic / lattice temperatures, as we will discuss in Reply 5).

According to the above discussion, we revised the sentences below in line 1, page 7.

In $F < F_c$, as already mentioned, the k_F oscillation as well as the anomaly in the xz orbital response disappear, and τ_1^{-1} becomes constant. In the XY Raman spectrum, the critical T -linear behavior was found in the inverse of the QEP intensity above T_s (Ref. 15). By the detailed analysis of the Curie-Weiss-like T -dependent nematic susceptibility in the form of $|T - T_0|^{-1}$,¹⁵ the authors derived the bare electronic nematic transition temperature T_0 that should describe the ideal nematic transition purely driven by electrons without any influence of lattice. For FeSe, T_0 was estimated to be far below T_s , *i.e.* 8 K, 20 K (Ref.15) and 30 K (Ref.31). The critical behavior of t_p^{-1} and τ_1^{-1} toward $F \approx 40 \pm 20 \mu\text{Jcm}^{-2}$, *i.e.* much smaller than F_c , may be reflecting that the base temperature of the TARPES measurements (20 K) is close to T_0 . This scenario is also consistent with the initial photo-response of Δk_{Fy} with small threshold ($< 30 \mu\text{Jcm}^{-2}$, see Fig. 2c). These results indicate that the electronic nematicity in the initial ultrafast regime (~ 120 fs) shows the flexible photo-reaction by decoupling from the lattice.

4) Another minor question is related to the meaning of k_F in non-equilibrium. The electronic distribution function is clearly broadened and may not even be thermal. Looking at Fig.1(b), it looks as if there should be a substantial (systematic) error bar being associated with extracting k_F in non-equilibrium, but the data in panel (d) contain no error bars.

We thank for the Reviewer's important advice. As he/she mentioned, we are measuring the electronic state in non-equilibrium. However, in our time-region of interest ($t > 120$ fs), the electronic distribution obeys the Fermi-Dirac function with the elevated electronic temperature [will be discussed in 5)]. In this situation, we can safely obtain k_F by fitting the momentum distribution curves at E_F . Regarding the errors, we fully agree that we should clearly indicate the error bars of Δk_{Fy} in Fig. 2d. We apologize for missing this information. Here we added the error bars estimated from the fitting analysis of the MDCs using a Lorentz function as shown in Fig. R1. For example, the error bars for $F = 430 \mu\text{J/cm}^2$ were estimated to be $\pm 0.0011 \text{ \AA}^{-1}$ at $t = -400$ fs, $\pm 0.0017 \text{ \AA}^{-1}$ at $t = 120$ fs and $\pm 0.0013 \text{ \AA}^{-1}$ at $t = 4000$ fs, respectively. As can be seen, this modification does not severely affect the context of the manuscript.

Fig. R1 F dependence of $\Delta k_{Fy}(t)$ obtained at 20 K with error bars.

According to this modification, we modified Fig. 2d in the revised manuscript as follows.

5) It would be useful if authors try to estimate the hypothetical final temperature of the electronic system based on the energy they deposit into the system, and compare this with T_S .

We thank for the Reviewer’s valuable suggestion. To answer this important issue, here we directly estimate the electronic temperature (T_e) from the fitting analysis of the momentum-integrated EDCs. In general, T_e should be estimated by using the momentum-integrated EDC spectrum which represents the total density of states multiplied by the Fermi-Dirac function further convoluted by the instrumental resolution function, as performed in the previous TARPES [*PRB* **89**, 115126 (2014)]. We integrated the EDCs of ARPES on xz from 0.0 \AA^{-1} to 0.17 \AA^{-1} along k_y , and fitted by a FD function convoluted by the gaussian of energy resolution (20 meV), assuming the constant density of states near E_F (Fig. R2a). After the photoexcitation of $220 \mu\text{Jcm}^{-2}$, T_e reaches $88 \pm 2 \text{ K}$ at 120 fs. Then, it shows a rapid decrease in $< 1 \text{ ps}$ and remains nearly constant at 45 K for $t > 3000 \text{ fs}$ (Fig. R2b), which is considerably lower than $T_s = 90 \text{ K}$.

According to the two-temperature model [*J. Exp. Theor. Phys.* **66**, 375 (1974)], elevated T_e approaches a constant value after the rapid relaxation *via* the electron-lattice coupling. There, the quasi-equilibrium state is realized, where the temperatures of electrons and lattice become equivalent. This behavior has been indeed discussed in the ultrafast optical measurements of the iron-based superconductors [*Nature Communications* **5**, 3229 (2014)]. The maximum lattice temperature is thus expected to be $\sim 45 \text{ K}$ in the present TARPES case. These analyses suggest that the electronic nematic order gets dissolved in the ultrafast regime while the lattice well maintains the orthorhombicity. With this, one of our main conclusion “*the electronic nematic dynamics is uncoupled from the lattice*” is well supported.

Fig. R2 **a**, Transient EDCs for xz and FD functions assuming a constant density of state. **b**, Time dependence of the electronic temperature for xz at $220 \mu\text{J}/\text{cm}^2$.

According to the new data analysis regarding the estimation of T_e , we added the discussion below in line 11, page 7.

Our analysis on the transient electronic temperature (T_e) (Supplementary section 7) indeed shows that T_e immediately reaches 88 ± 2 K at 120 fs and then decreases in less than 1 ps (Fig. S6a, b). For $t > 3$ ps, it becomes nearly constant at ~ 45 K, indicating the realization of quasi-equilibrium state where the temperatures of electrons and lattice become equivalent through the electron-lattice coupling³². The maximum lattice temperature is thus much lower than T_s ($= 90$ K), showing that the lattice stays orthorhombic.

We also added the reference number 32.

[32] Anisimov, S. I. *et al.*, Electron emission from metal surfaces exposed to ultrashort laser pulses. *J. Exp. Theor. Phys.* **66**, 375 (1974).

According to this modification, the discussion below in line 24, on page 6 of the previous manuscript was removed.

Being at $T \approx T_0$, the nematic FS may lose its C_2 property even by a weak photo-excitation that never raises the electron temperature close to T_s , where the nematicity starts to decrease in thermal equilibrium (19). If this is the case, the electronic nematicity at least in this initial ultrafast regime (~ 120 fs) should be decoupled from the lattice.

We also added the supplementary section 7 with the supplementary Fig. S6 as following.

Section S7. Estimation of the transient electronic temperature.

Here we estimate the electronic temperature (T_e) from the fitting analysis of the momentum-integrated EDCs. In general, T_e should be estimated by using the momentum-integrated EDC spectrum which represents the total density of states multiplied by the Fermi-Dirac function further convoluted by the instrumental resolution function. We integrated the EDCs of ARPES on xz from 0.0 \AA^{-1} to 0.17 \AA^{-1} along k_y , and fitted by a FD function convoluted by the gaussian of energy resolution (20 meV), assuming the

constant density of states near E_F (Fig. S6a). After the photoexcitation of $220 \mu\text{Jcm}^{-2}$, T_e reaches $88 \pm 2 \text{ K}$ at 120 fs. Then, it shows a rapid decrease in $< 1 \text{ ps}$ and remains nearly constant at 45 K for $t > 3000 \text{ fs}$ (Fig. S6b), which is considerably lower than $T_s = 90 \text{ K}$.

According to the two-temperature model^{S5}, elevated T_e approaches a constant value after the rapid relaxation *via* the electron-lattice coupling. There, the quasi-equilibrium state is realized, where the temperatures of electrons and lattice become equivalent. This behavior has been indeed discussed in the ultrafast optical measurements of the iron-based superconductors^{S6}. The maximum lattice temperature is thus expected to be $\sim 45 \text{ K}$ in the present TARPES case. These analyses suggest that the electronic nematic order gets dissolved in the ultrafast regime while the lattice well maintains the orthorhombicity.

Fig. S6 **Estimation of the electronic temperature.** **a**, Transient EDCs for xz and FD functions assuming a constant density of states. **b**, Time dependence of the electronic temperature for xz at $220 \mu\text{J}/\text{cm}^2$.

We also added the references below in the supplementary information.

[S5] Anisimov, S.I. *et al.*, *J. Exp. Theor. Phys.* **66**, 375 (1974).

[S6] Patz, A. *et al.*, *Nature Communications* **5**, 3229 (2014).

Reply to the Reviewer #2 :

The authors present a time-resolved ARPES study of detwinned FeSe, and report on the topic of pump-probe effects of electronic nematicity in this material. This is potentially an interesting study on the out-of-equilibrium effects of electronic nematicity, which would be novel to the field. However, there are several issues that prevent me from recommending publication in its current form.

We sincerely thank Reviewer 2 for evaluating our work and giving us valuable comments. We have now answered all the comments from the Reviewer and added new section of the Supplementary information according to his/her advice.

1. Could the authors comment on the effect of transient heating? Since the pump pulse necessarily deposits energy unto the sample, which very likely heats the sample up, it is hard to disentangle whether the observed disappearance of nematicity is due to intrinsic excitation out of this electronic phase at a temperature below T_S or simply the effect of having raised the electronic temperature to above T_S in this material. The transient heating would also have a fluence dependence where beyond a critical fluence the sample would always be pumped above T_S , and a saturation of the change in k_F is observed. This is an important issue regarding the argument that the pumped response probes the electronic nematicity decoupled from the lattice degree of freedom because in the case of transient heating the disappearance of electronic nematicity is observed on a tetragonal lattice via raised temperature, and in the case without transient heating the observed disappearance of electronic nematicity is observed on an orthorhombic lattice at base temperature.

We thank for the Reviewer's valuable suggestion. We agree that clarifying the effect of the transient heating is an important issue for correctly understanding the present TARPES results. For discussing the transient heating, it is important to introduce the temperatures for the electronic system (T_e) and the lattice system (T_l). In the present case, we can conclude that T_e reaches $\sim T_s$ ($= 90$ K) in the ultrafast regime when strongly excited, while T_l is always considerably lower than T_s (i.e. the lattice constantly remains orthorhombic). This indicates that the electrons are instantaneously decoupled from the lattice, giving rise to the nearly pure dynamics of the many-body electrons system. This is also consistent with the very small F threshold ($\ll F_c$) behavior of Δk_F as seen in Fig. 2c. The estimation of T_e is described in the following.

The transient heating should raise both the electronic temperature (T_e) and lattice temperature (T_l) after photoexcitation. According to the two-temperature model [*J. Exp. Theor. Phys.* **66**, 375 (1974)] which has been widely applied to the ultrafast dynamics in iron-based superconductors [*Nature Communications* **5**, 3229 (2014)], the responses of T_e and T_l are not equivalent until they reach the quasi-

equilibrium state through the energy transfer from the electrons to the lattice. This process usually takes few ps depending on the strength of the electron-phonon coupling.

For understanding the transient heating effect, here we directly estimate the electronic temperature (T_e) from the fitting analysis of the momentum-integrated EDCs. In general, T_e should be estimated by using the momentum-integrated EDC spectrum which represents the total density of states multiplied by the Fermi-Dirac function further convoluted by the instrumental resolution function, as performed in the previous TARPES [*PRB* **89**, 115126 (2014)]. We integrated the EDCs of ARPES on xz from 0.0 \AA^{-1} to 0.17 \AA^{-1} along k_y , and fitted by a FD function convoluted by the gaussian of energy resolution (20 meV), assuming the constant density of states near E_F (Fig. R1a). After the photoexcitation of $220 \mu\text{Jcm}^{-2}$, T_e reaches $88 \pm 2 \text{ K}$ at 120 fs. Then, it shows a rapid decrease in $< 1 \text{ ps}$ and remains nearly constant at 45 K for $t > 3000 \text{ fs}$ (Fig. R1b), which is considerably lower than $T_s = 90 \text{ K}$.

According to the two-temperature model [*J. Exp. Theor. Phys.* **66**, 375 (1974)], elevated T_e approaches a constant value after the rapid relaxation *via* the electron-lattice coupling. There, the quasi-equilibrium state is realized, where the temperatures of electrons and lattice become equivalent. This behavior has been indeed discussed in the ultrafast optical measurements of the iron-based superconductors [*Nature Communications* **5**, 3229 (2014)]. The maximum lattice temperature is thus expected to be $\sim 45 \text{ K}$ in the present TARPES case. These analyses suggest that the electronic nematic order gets dissolved in the ultrafast regime while the lattice well maintains the orthorhombicity. We also note that the orthorhombicity requires the longer time and higher pump fluence to be fully suppressed. According to the pump-probe X-ray diffraction measurement on BaFe_2As_2 , 10 % suppression of the orthorhombicity occurs in 30 ps by $F = 3.3 \text{ mJ/cm}^2$ [*Struct. Dyn.* **3**, 023611 (2016)].

Owing to the Reviewer's valuable advice, we believe that one of the main claims "***the pumped response probes the electronic nematicity decoupled from the lattice degree of freedom***" has become much clearer.

Fig. R1 **a**, Transient EDCs for xz and FD functions assuming a constant density of states. **b**, Time dependence of the electronic temperature for xz at $220 \mu\text{J}/\text{cm}^2$.

According to the new data analysis regarding the estimation of T_e , we added the discussion below in line 11, page 7.

Our analysis on the transient electronic temperature (T_e) (Supplementary section 7) indeed shows that T_e immediately reaches 88 ± 2 K at 120 fs and then decreases in less than 1 ps (Fig. S6a, b). For $t > 3$ ps, it becomes nearly constant at ~ 45 K, indicating the realization of quasi-equilibrium state where the temperatures of electrons and lattice become equivalent through the electron-lattice coupling³². The maximum lattice temperature is thus much lower than T_s ($= 90$ K), showing that the lattice stays orthorhombic.

We also added the reference number 32.

[32] Anisimov, S. I. *et al.*, Electron emission from metal surfaces exposed to ultrashort laser pulses. *J. Exp. Theor. Phys.* **66**, 375 (1974).

According to this modification, the discussion below in line 24, on page 6 of the previous manuscript was removed.

Being at $T \approx T_0$, the nematic FS may lose its C_2 property even by a weak photo-excitation that never raises the electron temperature close to T_s , where the nematicity starts to decrease in thermal equilibrium (19). If this is the case, the electronic nematicity at least in this initial ultrafast regime (~ 120 fs) should be decoupled from the lattice.

We also added the supplementary section 7 with the supplementary Fig. S6 below.

Section S7. Estimation of the transient electronic temperature.

Here we estimate the electronic temperature (T_e) from the fitting analysis of the momentum-integrated EDCs. In general, T_e should be estimated by using the momentum-integrated EDC spectrum which represents the total density of states multiplied by the Fermi-Dirac function further convoluted by the instrumental resolution function. We integrated the EDCs of ARPES on xz from 0.0 \AA^{-1} to 0.17 \AA^{-1} along k_y , and fitted by a FD function convoluted by the gaussian of energy resolution (20 meV), assuming the constant density of states near E_F (Fig. S6a). After the photoexcitation of $220 \mu\text{Jcm}^{-2}$, T_e reaches 88 ± 2 K at 120 fs. Then, it shows a rapid decrease in < 1 ps and remains nearly constant at 45 K for $t > 3000$ fs (Fig. S6b), which is considerably lower than $T_s = 90$ K.

According to the two-temperature model^{S5}, elevated T_e approaches a constant value after the rapid relaxation *via* the electron-lattice coupling. There, the quasi-equilibrium state is realized, where the temperatures of electrons and lattice become equivalent. This behavior has been indeed discussed in the ultrafast optical measurements of the iron-based superconductors^{S6}. The maximum lattice temperature is thus expected to be ~ 45 K in the present TARPES case. These analyses suggest that the electronic nematic order gets dissolved in the ultrafast regime while the lattice well maintains the orthorhombicity.

Fig. S6 **Estimation of the electronic temperature.** **a**, Transient EDCs for xz and FD functions assuming a constant density of states. **b**, Time dependence of the electronic temperature for xz at $220 \mu\text{J}/\text{cm}^2$.

We also added the references below in the supplementary information.

[S5] Anisimov, S.I. *et al.*, *J. Exp. Theor. Phys.* **66**, 375 (1974).

[S6] Patz, A. *et al.*, *Nature Communications* **5**, 3229 (2014).

2. Going along with the last point, one way to disentangle this is perhaps to look at the EDCs after pumping, such as in Fig. 3 and the supplement figures. Could the EDCs be fitted to a functional form with a Fermi-Dirac function convolved with the instrumental resolution to extract the electronic temperature as a function of delay time? If the change is due purely to electronic dynamics, one may expect the EDCs to show a peak or at least hump-like feature above E_F where the authors argue that the xz states lie, and where the pump excites the states below E_F to. The EDCs in the figure currently look perhaps more like thermally broadened EDCs with a single peak below E_F . Could some fittings under the two scenarios resolve this issue?

We thank the Reviewer for his/her valuable suggestion. As replied to the Comment 1, we analyzed the momentum-integrated EDC of xz using a Fermi Dirac function convoluted with the instrumental resolution. Regarding the shape of EDCs, we can conclude that they look like the thermally broadened spectra with elevated T_e , at least in the $t > 120$ fs region where we discuss the nematic dynamics in this paper. This is

because the electrons usually get quickly thermalized by the electron-electron interaction, typically in 10-100 fs. The peak/ hump structure above E_F should definitely appear in a more ultrafast regime (e.g. $< \sim 10$ fs), especially when much shorter optical pulses are used (in the present work, the pulse duration is 250 fs).

3. Along the same lines, comparing Fig. 3a and 3b, if the pumping excites xz electrons to unoccupied xz states above E_F , where do the pumped yz electrons go in panel b? It seems that there is substantial spectral weight depletion after pumping in the yz states. Then where does it go if it cannot go into the unoccupied xz states?

We thank the Reviewer for his/her valuable comment. As the Reviewer pointed out, we find substantial spectral weight depletion in the yz states after pumping. While it awaits further investigations, we consider that the photo-excited yz electrons may be partly trapped at the M point. Figure R2 shows the calculated band structure of FeSe in the tetragonal state [*J. Appl. Phys.* **115**, 193907 (2014)]. Because of the semi-metallic electronic structure i.e. hole bands at Γ and electron bands at M, a part of electrons excited by 1.5 eV photons at Γ may quickly gather around the electron bands at M in the relaxation process (red arrow in Fig. R2). Furthermore, the sign of the orbital polarization is opposite between Γ and M in FeSe [*PRB*, **92**, 205117 (2015)], which realizes the higher occupation of the yz electrons near E_F at M. Considering the Dirac-cone-like band crossing just below E_F at M [*PRB*, **90**, 121111 (2014)], the carrier relaxation may have a bottle neck which will elongate the depletion of the yz electrons at Γ . To fully clarify the carrier dynamics, the high photon-energy TARPES measurement to access a wider momentum region and higher energy above E_F is necessary.

Fig. R2 Calculated band structure for FeSe in the tetragonal state. [*J. Appl. Phys.* **115**, 193907 (2014).]

We also modified the sentence in line 4 page 5 of previous manuscript as following.

As shown in Fig. 3g,h, $\Delta I(t)$ of xz exhibits the retarded maximum at $t_{\text{ret}} = \sim 700$ fs, whereas the yz electrons show the simple exponential decay similar to the weak-excitation regime, with the initial maximum at ~ 250 fs.

→

As shown in Fig. 3g,h, $\Delta I(t)$ of xz exhibits the retarded maximum at $t_{\text{ret}} = \sim 700$ fs, whereas the yz electrons show the exponential decay more or less similar to the weak-excitation regime, with the initial maximum at ~ 250 fs.

4. There is an oscillatory component in the delay response for higher fluences. This, if real, would not be due to transient heating. Could the authors discuss more the potential origin of this mode?

We thank the Reviewer for his/her important comment. As the Reviewer pointed out, the discussion on the potential origin of the oscillatory behavior in Δk_{Fy} was rather inconclusive. The most important aspect is that the observed anti-phase oscillation of Δk_{Fx} and Δk_{Fy} , as shown in Fig. 2f, is the direct representation of Pomeranchuk-type Fermi surface oscillation. To make this point clear, we added the description below in line 8, page 4.

The observed anti-phase oscillation of k_{Fx} and k_{Fy} rather directly represents the Pomeranchuk-type oscillation of FS²⁹, being intensively discussed as the fundamental excitation in the electronic nematic state.

Accordingly, we added new reference number 29 as following.

[29] Pomeranchuk, I. Ia. On the stability of a Fermi liquid. *J. Exp. Theor. Phys.* **8**, 361 (1959)

In addition, we modify the discussion part. As described in the previous manuscript, electronic Raman scattering measurements [PNAS **113**, 9177 (2016)] reported the critical behavior in the nematic susceptibility with the XY -symmetry for FeSe. By further referring to another recent study [arXiv:1710.09892], we can more deeply discuss the similarities between the nematic dynamics obtained by the electronic Raman scattering and our TARPES. In the Raman scattering study, the critical enhancement of nematic susceptibility and corresponding quasi-elastic peak (QEP) in the XY spectrum are observed in the tetragonal phase, on cooling toward the structural transition temperature (T_s). It implies the presence of the critical nematic fluctuation in $T > T_s$. On cooling below $T < T_s$, on the other hand, the QEP Raman intensity rapidly diminishes, and a gap opens in the XY spectrum indicating the sudden suppression of low-energy excitations [Fig. 1(b) in arXiv:1710.09892]. On further cooling below the superconducting transition temperature (T_c), they find a peak appearing at 3.6 meV, which they assign to the nematic resonance mode acquiring the coherence in the superconducting state. First we note that the energy of this nematic mode (3.6 meV) is fairly close to that of the Pomeranchuk-type FS oscillation (3.1 meV for 220 $\mu\text{J}/\text{cm}^2$) obtained by TARPES, thus suggesting the similarity in the origin. Nevertheless, we have to also note that our measurements are done in the non-superconducting state ($T \sim 20 \text{ K} > T_c$), where the coherent nematic mode does not exist. Our present interpretation is that the observed short-life FS oscillation should be associated with the QEP (i.e. nematic fluctuation) in the Raman study, since the T -dependence of QEP is very similar to the F -dependence of Δk_{FY} oscillation; apparent only in $F > F_c$, and suddenly disappears in $F < F_c$. The nematic fluctuation should be of course incoherent in nature, however, we consider that by instantaneously triggering the dissolution of the nematic state, it can appear as the heavily-damped oscillatory response in the non-equilibrium time domain.

With these facts, we believe that we can offer the specific microscopic picture of the nematic excitation / fluctuation (i.e. the Pomeranchuk-type FS oscillation accompanying the orbital redistribution), which has been long discussed in the community due to its ubiquitous nature, but without being well identified.

To account for the Reviewer's advices, we added the sentences below in line 6, page 6 of the revised manuscript.

The nematic-orbital excitation obtained in the present TARPES shows a striking resemblance with the nematic dynamics in thermal equilibrium as probed by the recent Raman scattering measurements.^{15,30} The electronic Raman spectra of XY symmetry (X and Y are coordinates along the crystal axes of the tetragonal setting) show the characteristic quasi-elastic peak (QEP) evolving toward T_s on cooling the temperature (T), discussed in terms of nematic susceptibility enhancement.^{15,30} The QEP rapidly diminishes at $T < T_s$, on the other hand, and a gap opens in the XY Raman spectra thus indicating the suppression of

low-energy nematic excitations (Ref. 30). These behaviors are reminiscent of the nematic-orbital excitation observed by TARPES, where the peculiar slowing behavior shows up in $F > F_c$, and the excitation itself suddenly disappears in $F < F_c$. The XY Raman spectrum further reveals a peak at 3.6 meV in the superconducting state ($T < T_c$), which is interpreted in Ref [30] as the nematic resonance mode acquiring the coherence by the superconducting gap opening. This energy scale is fairly close to that of the damped k_F oscillation (3.1 meV) observed by TARPES near F_c , thus suggesting the similarity in its origin. With these facts, we presently consider that the nematic-orbital excitation obtained by TARPES should be associated with the QEP (i.e. nematic fluctuation) in the Raman study. The nematic fluctuation is incoherent in nature, however, by instantaneously triggering the dissolution of the nematic state, it may be appearing as the heavily-damped oscillatory response in the time domain.

We also added the sentences below in line 1, page 7 of the revised manuscript.

In $F < F_c$, as already mentioned, the k_F oscillation as well as the anomaly in the xz orbital response disappear, and τ_1^{-1} becomes constant. In the B_{1g} Raman spectrum, the critical T -linear behavior was found in the inverse of the QEP intensity above T_s (Ref. 15). By the detailed analysis of the Curie-Weiss-like T -dependent nematic susceptibility in the form of $|T - T_0|^{-1}$,¹⁵ the authors derived the bare electronic nematic transition temperature T_0 that should describe the ideal nematic transition purely driven by electrons without any influence of lattice. For FeSe, T_0 was estimated to be far below T_s , i.e. 8 K, 20 K (Ref.15) and 30 K (Ref.31). The critical behavior of t_p^{-1} and τ_1^{-1} toward $F \approx 40 \pm 20 \mu\text{Jcm}^{-2}$, i.e. much smaller than F_c , may be reflecting that the base temperature of the TARPES measurements (20 K) is close to T_0 . This scenario is also consistent with the initial photo-response of Δk_{Fy} with small threshold ($< 30 \mu\text{Jcm}^{-2}$, see Fig. 2c). These results indicate that the electronic nematicity in the initial ultrafast regime (~ 120 fs) shows the flexible photo-reaction by decoupling from the lattice.

We also added the reference number 30 as follows.

[30] Zhang, W.-L. *et al.*, Stripe quadrupole order in the nematic phase of FeSe_{1-x}S_x. arXiv:1710.09892.

5. In ref. 25, where a similar tr-ARPES is done on FeSe films in the same fluence regime, a clear coherent oscillation is reported. But such oscillation is not seen in the current set of data. Could the authors clarify in the text the reason of this inconsistency? It is likely due to the different regimes of time/energy resolution used, but would be good for the general public to know.

We thank for the Reviewer's valuable suggestion. The absence of the signatures of the coherent optical phonons should be due to the lower time resolution in this study, as pointed out by the Referee. (Note that

since the time- and energy-resolutions are bound by the Heisenberg uncertainty principle, the energy-resolution is better in the present work.) The oscillation period of the A_{1g} optical phonon mode for FeSe is 190 fs [*Science* **357**, 71 (2017)]. Since this is considerably shorter than the pulse duration of the present TARPES measurements (250 fs), they seem to be severely suppressed.

To address this point clearly, we add the sentence below in line 6, page 4, as the footnote in reference number 28.

[28] We note that the previously reported A_{1g} optical phonons²⁵ are absent in the present TARPES data, because of the time resolution (250 fs).

6. Around the Gamma point, as has been reported, there is substantial spin-orbit coupling that mixes the xz and yz orbitals. How does this energy scale affect the dynamics of the pump-probe process (ie transitions between xz/yz states)?

We thank for the Reviewer's interesting comment. Spin-orbit interaction (SOI) in FeSe lifts the degeneracy between xz and yz orbital bands and makes the gap by 20 meV at Γ [*PRB* **92**, 205117 (2015)]. Since the energy gap between bands leads to the bottleneck of carrier relaxation process in general, the larger SOI gap may give rise to the slower carrier relaxation from α band to β band. The possible role of SOI on the yz vs xz orbital redistribution, however, is still unclear at present. It should be further clarified in future, mainly from the theoretical aspect.

Reply to the Reviewer #3:

The authors of the article performed time resolved ARPES experiments on FeSe de-twinned by external stress. They monitored the dynamics of electronic states in the nematic phase along the two high symmetry axis of the crystal. The data indicate that photoexcitation with moderate fluence can disrupt the nematic order on an ultrafast timescale. The melting and recovery time are comparable to the period of a coherent modulation appearing at intermediate fluence. This article is very interesting and will have high impact in the scientific community. It is clearly written and self-consistent. The large amount of experimental data that are discussed will provide an important reference for future works.

We thank the Reviewer 3 for highly evaluating our work and recommending the publication in Nature Communications. We have now answered all the comments from the Reviewer.

My only criticism regards the speculative attempt to identify the fluence dependence of the timescales with a critical behavior. Could the author explain what is a quantum critical point? A hypothetical nematic phase that, if decoupled from the lattice, would display a transition temperature of 20 K does not lead to a quantum critical point. On which basis such hypothetical phase would lead to fluctuations affecting linearly the electronic timescale if the fluence overcomes the threshold value?

First we would like to apologize that the statement on the quantum critical point at the last part (line 23, page 6 in the previous version) was perhaps misleading. As he/she pointed out, FeSe shows the nematic phase transition at $T_s = 90$ K ($T_0 = 20$ K for the hypothetical electronic phase), which is not located at the quantum critical point (QCP). Nevertheless, we believe that the critical behavior of various timescales in the fluence dependence can be discussed in analogy with those appearing in the vicinity of temperature- or any parameter-driven (e.g. pressure, magnetic/electric field, etc) phase transitions. In general, when there is a phase transition at the critical parameter of P_c , the critical phenomenon appears in many physical quantities e.g. susceptibility, correlation length, specific heat, and so on, in the $|P - P_c|^a$ form (a : critical exponent). In the present case of FeSe, as reported by Raman scattering studies, the intensity of the quasi-elastic peak (interpreted as the nematic susceptibility) shows the $|T - T_0|^{-1}$ behavior at $T > T_s$, thus indicating the critical fluctuation enhancement toward T_0 ($\ll T_s$), which quenches below the structural transition T_s [*PNAS* **113**, 9177 (2016)]. The F -dependent time-scales of damped k_F oscillation as well as the xz orbital retardation are reminiscent of such behavior, i.e. $|F - F_0|^{-1}$ property with $F_0 = 40 \pm 20 \mu\text{Jcm}^{-2}$ ($\ll F_c = 220 \mu\text{Jcm}^{-2}$) at $F > F_c$, which suddenly disappears in $F < F_c$. Considering this striking similarity, we conclude

that the heavily damped k_F oscillation and the xz orbital retardation are associated with the critical nematic fluctuation observed by Raman studies.

Regarding the quantum criticality, we can access the QCP of the nematic phase by replacing 17% of Se with S in Fe(Se,S) system [*PNAS* **113**, 8139 (2016)]. It will be very interesting to study this QCP by TARPES. Near the QCP, the electronic Raman scattering shows a drastic enhancement of the nematic (quantum) fluctuations toward 0 K. In this case, the FS dynamics should be dominated by the enhanced nematic fluctuation also in the very weakly excited conditions, and the threshold fluence F_c is expected to approach zero. For clarifying the dynamics of the electronic nematicity near the QCP, further TARPES measurements on the FeSe_{1-x}S_x system are highly desired.

To account for this important issue, we changed the sentence below in line 1, page 8.

“The ultrafast photo-excitation adds a new possibility of understanding and manipulating the itinerant electron systems near the quantum critical point, associated with unprecedented phenomena such as exotic superconductivity, peculiar magnetism, thermopower enhancement, and so on.”

→

“Further studies on the nematic dynamics around the quantum critical point in FeSe_{1-x}S_x system³¹ and the coherent nematic resonance mode in the superconducting state are highly desired. The ultrafast photo-excitation adds a new possibility of understanding and manipulating the large-amplitude electronic fluctuations associated with unprecedented phenomena such as exotic superconductivity, peculiar magnetism, thermopower enhancement, and so on.”

Summary of changes;

1. We changed the word “is” into “are” in line 1, page 3.
2. We modified the equation in line 3, page 4 as follows.

$$k_F(t) = k_{F1}\exp(-t/\tau_1) + k_{F2}\exp(-t/\tau_2) + k_{F3}\exp(-t/\tau_3)\cos(2\pi t/t_p)$$

→

$$k_F(t) = k_{F0} + k_{F1}\exp(-t/\tau_1) + k_{F2}\exp(-t/\tau_2) + k_{F3}\exp(-t/\tau_3)\cos(2\pi t/t_p)$$

3. We added the words “attributed to” in line 6, page 4.
4. We added the footnote below in line 6, page 4, as denoted by new reference number 28.

[28] We note that the previously reported A_{1g} optical phonons²⁵ are absent in the present TARPES data, because of the time resolution (250 fs).

5. We added the words “(~22 meV)” in line 7, page 4.
6. We modified the words “3 meV for 220 μJcm^{-2} ” into “3.1 meV for 220 μJcm^{-2} ” in line 7, page 4.
7. We changed the sentence in line 8, page 4 in the previous manuscript as following.

It should be also distinguished from amplitude modes of symmetry-broken states^{22,28} (i.e. nematic-order in this case), considering that the oscillations are rather lacking in the ordered weak-excitation regime.

→

The observed anti-phase oscillation of k_{Fx} and k_{Fy} rather directly represents the Pomeranchuk-type oscillation of FS²⁹, being intensively discussed as the fundamental excitation in the electronic nematic state.

8. We removed the previous reference number 28 and added new reference number 29.

[29] Pomeranchuk, I. Ia. On the stability of a Fermi liquid. *J. Exp. Theor. Phys.* **8**, 361 (1959).

9. We modified the sentence in line 4 page 5 of previous manuscript as following.

As shown in Fig. 3g,h, $\Delta I(t)$ of xz exhibits the retarded maximum at $t_{\text{ret}} = \sim 700$ fs, whereas the yz electrons show the simple exponential decay similar to the weak-excitation regime, with the initial maximum at ~ 250 fs.

→

As shown in Fig. 3g,h, $\Delta I(t)$ of xz exhibits the retarded maximum at $t_{\text{ret}} = \sim 700$ fs, whereas the yz electrons show the exponential decay more or less similar to the weak-excitation regime, with the initial maximum at ~ 250 fs.

10. We modified the words, “orbital flipping” into “retarded maximum in xz component” in line 18, page 5.

11. We added the word “similarly” in line 18, page 5.

12. We modified the words, “ yz -to- xz orbital-flipping process” into “orbital-dependent carrier dynamics” in line 20, page 5.

13. We added the sentences below in line 21 page 5 of the revised manuscript.

We note that the transient FS at $t_p/2$ ($\approx t_{\text{ret}}$) is more elliptical than that expected without the oscillatory response. Such an overshoot of the nematicity in FS should also appear in the orbital-dependent carrier dynamics. In the process relaxing back from C_4 isotropic to C_2 nematic ground state, the electrons at the band top (black rectangle in Fig. 1 b,d) change their orbital characters from “(nearly) xz/yz degenerate” to “predominantly xz ”. The retarded maximum in $I(t)$ for xz can be thus regarded as an indication of the *orbital redistribution* from yz to xz (Fig. 4a). The synchronized responses in the FS oscillation and orbital-dependent carrier dynamics thus represent the *nematic-orbital excitation*.

14. We added the sentences below in line 6, page 6 of the revised manuscript.

The nematic-orbital excitation obtained in the present TARPES shows a striking resemblance with the nematic dynamics in thermal equilibrium as probed by the recent Raman scattering measurements.^{15,30} The electronic Raman spectra of XY symmetry (X and Y are coordinates along the crystal axes of the tetragonal setting) show the characteristic quasi-elastic peak (QEP) evolving toward T_s on cooling the temperature (T), discussed in terms of nematic susceptibility enhancement.^{15,30} The QEP rapidly diminishes at $T < T_s$, on the other hand, and a gap opens in the XY Raman spectra thus indicating the suppression of low-energy nematic excitations (Ref. 30). These behaviors are reminiscent of the nematic-orbital excitation

observed by TARPES, where the peculiar slowing behavior shows up in $F > F_c$, and the excitation itself suddenly disappears in $F < F_c$. The XY Raman spectrum further reveals a peak at 3.6 meV in the superconducting state ($T < T_c$), which is interpreted in Ref [30] as the nematic resonance mode acquiring the coherence by the superconducting gap opening. This energy scale is fairly close to that of the damped k_F oscillation (3.1 meV) observed by TARPES near F_c , thus suggesting the similarity in its origin. With these facts, we presently consider that the nematic-orbital excitation obtained by TARPES should be associated with the QEP (i.e. nematic fluctuation) in the Raman study. The nematic fluctuation is incoherent in nature, however, by instantaneously triggering the dissolution of the nematic state, it may be appearing as the heavily-damped oscillatory response in the time domain.

15. We modified the figure “~3 meV” into “3.1 meV” in line 1, page 7.

16. We revised the sentences below in line 1, page 7:

In $F < F_c$, as already mentioned, the k_F oscillation as well as the anomaly in the xz orbital response disappear, and τ_1^{-1} becomes constant. In the B_{1g} Raman spectrum, the critical T -linear behavior was found in the inverse of the QEP intensity above T_s (Ref. 15). By the detailed analysis of the Curie-Weiss-like T -dependent nematic susceptibility in the form of $|T - T_0|^{-1}$,¹⁵ the authors derived the bare electronic nematic transition temperature T_0 that should describe the ideal nematic transition purely driven by electrons without any influence of lattice. For FeSe, T_0 was estimated to be far below T_s , i.e. 8 K, 20 K (Ref.15) and 30 K (Ref.31). The critical behavior of t_p^{-1} and τ_1^{-1} toward $F \approx 40 \pm 20 \mu\text{Jcm}^{-2}$, i.e. much smaller than F_c , may be reflecting that the base temperature of the TARPES measurements (20 K) is close to T_0 . This scenario is also consistent with the initial photo-response of Δk_{Fy} with small threshold ($< 30 \mu\text{Jcm}^{-2}$, see Fig. 2c). These results indicate that the electronic nematicity in the initial ultrafast regime (~ 120 fs) shows the flexible photo-reaction by decoupling from the lattice.

17. We added the error bars in Fig. 2d.

18. We removed the sentences in line 9, page 6 in the previous manuscript.

Being at $T \approx T_0$, the nematic FS may lose its C_2 property even by a weak photo-excitation that never raises the electron temperature close to T_s , where the nematicity starts to decrease in thermal equilibrium (19). If this is the case, the electronic nematicity at least in this initial ultrafast regime (~ 120 fs) should be decoupled from the lattice.

19. We added the discussion below in line 11, page 7.

Our analysis on the transient electronic temperature (T_e) (Supplementary section 7) indeed shows that T_e immediately reaches 88 ± 2 K at 120 fs and then decreases in less than 1 ps (Fig. S6a, b). For $t > 3$ ps, it becomes nearly constant at ~ 45 K, indicating the realization of quasi-equilibrium state where the temperatures of electrons and lattice become equivalent through the electron-lattice coupling³². The maximum lattice temperature is thus much lower than T_s ($= 90$ K), showing that the lattice stays orthorhombic.

20. We modified the words, “orbital flipping” into “orbital redistribution” in line 21, page 7.

21. We modified the sentences below in line 1, page 8.

“The ultrafast photo-excitation adds a new possibility of understanding and manipulating the itinerant electron systems near the quantum critical point, associated with unprecedented phenomena such as exotic superconductivity, peculiar magnetism, thermopower enhancement, and so on.”

→

“Further studies on the nematic dynamics around the quantum critical point in FeSe_{1-x}S_x system³¹ and the coherent nematic resonance mode in the superconducting state are highly desired. The ultrafast photo-excitation adds a new possibility of understanding and manipulating the large-amplitude electronic fluctuations associated with unprecedented phenomena such as exotic superconductivity, peculiar magnetism, thermopower enhancement, and so on.”

22. We also added the references as follows.

[30] Zhang, W.-L. *et al.*, Stripe quadrupole order in the nematic phase of FeSe_{1-x}S_x. arXiv:1710.09892.

[32] Anisimov, S. I. *et al.*, Electron emission from metal surfaces exposed to ultrashort laser pulses. *J. Exp. Theor. Phys.* **66**, 375 (1974).

23. We added the supplementary section 7 with the supplementary Fig. S6 as following.

Section S7. Estimation of the transient electronic temperature.

Here we estimate the electronic temperature (T_e) from the fitting analysis of the momentum-integrated EDCs. In general, T_e should be estimated by using the momentum-integrated EDC spectrum which represents the total density of states multiplied by the Fermi-Dirac function further convoluted by the instrumental resolution function. We integrated the EDCs of ARPES on xz from 0.0 \AA^{-1} to 0.17 \AA^{-1} along k_y , and fitted by a FD function convoluted by the gaussian of energy resolution (20 meV), assuming the constant density of states near E_F (Fig. S6a). After the photoexcitation of $220 \mu\text{Jcm}^{-2}$, T_e reaches $88 \pm 2 \text{ K}$ at 120 fs. Then, it shows a rapid decrease in $< 1 \text{ ps}$ and remains nearly constant at 45 K for $t > 3000 \text{ fs}$ (Fig. S6b), which is considerably lower than $T_s = 90 \text{ K}$.

According to the two-temperature model^{S5}, elevated T_e approaches a constant value after the rapid relaxation *via* the electron-lattice coupling. There, the quasi-equilibrium state is realized, where the temperatures of electrons and lattice become equivalent. This behavior has been indeed discussed in the ultrafast optical measurements of the iron-based superconductors^{S6}. The maximum lattice temperature is thus expected to be $\sim 45 \text{ K}$ in the present TARPES case. These analyses suggest that the electronic nematic order gets dissolved in the ultrafast regime while the lattice well maintains the orthorhombicity.

Fig. S6 **Estimation of the electronic temperature.** **a**, Transient EDCs for xz and FD functions assuming a constant density of states. **b**, Time dependence of the electronic temperature for xz at $220 \mu\text{J}/\text{cm}^2$.

24. We added the references below in the supplementary information.

[S5] Anisimov, S.I. *et al.*, *J. Exp. Theor. Phys.* **66**, 375 (1974).

[S6] Patz, A. *et al.*, *Nature Communications* **5**, 3229 (2014).

Reviewers' comments:

Reviewer #1 (Remarks to the Author):

As said in my previous report, the manuscript "Ultrafast nematic-orbital excitation in FeSe" contains an interesting and thorough experimental study of the dynamics of electronic nematicity after photoexcitation at optical frequencies in FeSe. My main concern was regarding the analysis, modeling and interpretation of the experimental data and which new physical insights are obtained from this study. The revised manuscript addresses most of my concerns, in particular regarding the analysis part.

The revised manuscript now includes results of the transient electronic "temperature" after photoexcitation (extracted from fitting EDCs for xz orbitals). This confirms that the lattice remains below the nematic transition T_s while the electronic temperature reaches a temperature $T \approx T_s$ shortly after/during the pump (after 120 fs). At longer times, the electronic temperature saturates at $T=45$ K, which is well below T_s . Wording of the manuscript has also been improved at various locations, making the manuscript more readable. The figures contain a lot of information, but are well organized and clearly arranged.

The interpretation and modeling aspect of the work is certainly improved but still rather speculative. On the positive side, this may trigger further studies on the subject. Overall, I am not a strong advocate for publication of the manuscript in Nature Communication, but I still support it, because this TARPES study is a very useful addition to the literature on FeSe that will stimulate further research on the subject, both in and out of equilibrium.

Before publication, I urge the authors to revise the discussion about comparison to Raman study, in particular, regarding the comparison with Raman results in the superconducting state. While a comparison to equilibrium Raman studies is a good starting point, it is too simplistic to associate certain excitations they find with modes seen in Raman in equilibrium simply based on similar energy scales. The reason is that in the TARPES experiment, the system is far from equilibrium, where entirely new physics may appear that is absent in (or close to) equilibrium, where the Raman study is performed. Furthermore, various possible excitations are nearby in energy and interactions and nonlinearities cannot be neglected out of equilibrium. Phonons, both acoustic and optical, are also present (and nearby in energy) and their effect is not discussed in much detail in the manuscript. It is not convincing that the (oscillation) timescale t_p , which authors extract from the xz photoelectron dynamics in the normal state (above superconducting T_c), should be associated with the Raman resonance mode observed only in the superconducting state (analogously to the neutron resonance mode). This discussion should be substantially revised before publication.

Further, I would suggest that authors discuss the potential impact of phonon dynamics on their observations, in particular, given that the optical phonon frequency they mention (22 meV \approx 0.2 ps) is also in the range of timescales/energy scales under investigation. Since the orbital content of the Fermi pockets is linked to the state of the lattice, phonon excitations may strongly affect the electronic carrier distribution in n_{xz} and n_{yz} orbitals as well. Since the pump is in the optical range, it is expected that optical phonons will also be excited. As their timescales is comparable to the few ps timescales observed in the experiment, it is not obvious that their effect can be neglected.

Minor questions that the authors should consider are:

- 1) Which fluence F is used in Fig.2 (a, b)?
- 2) Would it be possible to add a y-axis scale to Fig. 2 (d)?

Reviewer #2 (Remarks to the Author):

The authors have answered all questions in detail and I am satisfied with the modifications. The manuscript has been much improved and I am happy to recommend its publication.

One comment regarding the asymmetry between xz and yz after pumping in the weak fluence regime (Fig. 3a and b). It seems that what the authors put in the response to be rather important, that there is an asymmetric scattering channel for xz and yz between Γ and M in the nematic state, which could be the reason for the missing yz spectral weight at Γ after pumping. At least this seems consistent with the equilibrium ARPES results (and perhaps even the STM result) on FeSe in that the electron pocket carrying xz at M has been missing while the observed electron pocket is dominated by yz . Could it be useful to include a sentence in the main text pointing out this difference of behavior between xz and yz , if not the potential interpretation, at least a statement of the experimental observation?

Reviewer #3 (Remarks to the Author):

This article represents the first clear signature of electronic oscillations related to a nematic phase. As explained by the author the monitored mode can be detected even if the nematic order does not hold long range coherence. The high quality data on detwinned crystals are hard to obtain and demand strong experimental effort. Being an expert in time resolved ARPES, I could say that this article is more interesting than several recent works published in Science or Nature and that target just the ultrafast community. I am satisfied by the reply of the authors to my comment about quantum criticality. However the final discussion of the article is too speculative. I support publication in Nature Comm. but I encourage the authors fix the following issues:

a) It would be good to state that the $1/|T-T_c|$ behavior is expected from Gaussian fluctuations and not does not provide information about the universality class and critical exponents.

a) The dynamical scaling observed when the system is weakly perturbed and the temperature approach T_c is conceptually different the one observed when approaching the fluence to a threshold value. The authors may rather consider non-equilibrium criticality to explain their data.

c) On the base of the data that I have seen, there is no reason to claim that the excitation of nematic order is decoupled to the lattice. On one hand, the conjecture of an underlying electronic transition at lower temperature is badly posed. Indeed, it is the relevant coupling to the lattice that defines the experimental T_c . On the other hand, the observed critical fluence is not particularly small and also corroborates a significant coupling to the lattice. As a consequence the authors should revise misleading statements as "These real-time observations reveal the nature electronic nematic excitation decoupled from the underlying lattice."

Reply to the Reviewer #1:

We sincerely thank Reviewer 1 for positively evaluating our manuscript and giving us important suggestions and comments. By answering to all his/her questions and comments as described below, we believe that the discussion part is improved.

Before publication, I urge the authors to revise the discussion about comparison to Raman study, in particular, regarding the comparison with Raman results in the superconducting state. While a comparison to equilibrium Raman studies is a good starting point, it is too simplistic to associate certain excitations they find with modes seen in Raman in equilibrium simply based on similar energy scales. The reason is that in the TARPES experiment, the system is far from equilibrium, where entirely new physics may appear that is absent in (or close to) equilibrium, where the Raman study is performed. Furthermore, various possible excitations are nearby in energy and interactions and nonlinearities cannot be neglected out of equilibrium. Phonons, both acoustic and optical, are also present (and nearby in energy) and their effect is not discussed in much detail in the manuscript. It is not convincing that the (oscillation) timescale t_p , which authors extract from the xz photoelectron dynamics in the normal state (above superconducting T_c), should be associated with the Raman resonance mode observed only in the superconducting state (analogously to the neutron resonance mode). This discussion should be substantially revised before publication. Further, I would suggest that authors discuss the potential impact of phonon dynamics on their observations, in particular, given that the optical phonon frequency they mention (22 meV \approx 0.2 ps) is also in the range of timescales/energy scales under investigation. Since the orbital content of the Fermi pockets is linked to the state of the lattice, phonon excitations may strongly affect the electronic carrier distribution in n_{xz} and n_{yz} orbitals as well. Since the pump is in the optical range, it is expected that optical phonons will also be excited. As their timescales is comparable to the few ps timescales observed in the experiment, it is not obvious that their effect can be neglected.

We agree that the present comparison between the TARPES and electronic Raman scattering is somewhat speculative, and more detailed discussion on non-equilibrium critical phenomenon is certainly necessary. Nevertheless, the theoretical basis for such discussion is still severely lacking, and developing it may be beyond the scope of this paper. We strongly hope our work will stimulate the future investigations.

According to the Reviewer's advice, we removed the sentences describing the direct comparison between the observed Fermi surface oscillation and the nematic collective Raman mode in the superconducting state (line 14, page 6 in the previous manuscript), and added some discussion to stress the above point (line 20, page 7 in the present manuscript) .

We removed the statement (line 14, page 6 in the previous manuscript):

“The XY Raman spectrum further reveals a peak at 3.6 meV in the superconducting state ($T < T_c$), which is interpreted in Ref [30] as the nematic resonance mode acquiring the coherence by the superconducting gap opening. This energy scale is fairly close to that of the damped k_F oscillation (3.1 meV) observed by TARPES near F_c , thus suggesting the similarity in its origin.”

We added the discussion (line 20, page 7 in the present manuscript):

“This behavior is seemingly related to the critical nematic fluctuation, nevertheless, future theoretical studies on non-equilibrium critical phenomena are highly necessary.”

We also agree with the possible importance of the optical / acoustic phonon dynamics in relation to the nematic-orbital excitations detected by TARPES. Regarding the coherent A_{1g} optical phonon mode (22 meV, 190 fs) as reported by several TARPES studies [Science **357**, 71 (2017).], however, it is fairly difficult to observe in the present experimental setup utilizing the pump pulse of ~170 fs FWHM. Since this pulse duration is very close to the time period of the A_{1g} optical phonon mode, the oscillatory response may be cancelled out even though the total time resolution itself is not so bad (250 fs). To fully understand the nematic excitation in association with phonons, more systematic time-resolved measurements on electrons and lattice (TARPES, ultrafast x-ray & electron diffraction, *etc*) with shorter pulses are highly desired. To discuss these points, we added some descriptions as follows.

We added some description on A_{1g} phonons (line 9, page 4):

“The time scale of the oscillatory response (1.4 ps, 3.1 meV) is considerably slow as compared to the coherent A_{1g} optical phonon (190 fs, 22 meV) which is known to strongly couple to the electronic state in this system²⁴⁻²⁶. Their potential interplay is unfortunately hidden in the present TARPES data, possibly due to the duration of the pump pulse (170 fs) comparable to the time period of A_{1g} mode (190 fs) that tends to vanish the coherent oscillation.”

We added some description on future perspective (line 2, page 8):

“Systematic time-resolved diffraction measurements will also help understanding the possible interplay among the nematic excitation and the optical / acoustic phonons.^{25, 34,}”

We accordingly added new reference #34 as follow.

[34] Nakamura, A. *et al.*, *Evaluation of photo-induced shear strain in monoclinic VTe₂ by ultrafast electron diffraction*. *Appl. Phys. Exp.* **11**, 092601 (2018).

Minor questions that the authors should consider are:

1) Which fluence F is used in Fig.2 (a, b)?

We thank for the Reviewer's advice. The fluence for the Fig. 2(a,b) is 220uJ/cm². We added the information in the caption of Fig. 2(a,b).

2) Would it be possible to add a y-axis scale to Fig. 2 (d)?

We thank for the Reviewer's suggestion. The y-axis scale bar was shown in Fig.2(d) but it was perhaps too thin. We added the zero label and y-axis scale for each curve.

Reply to the Reviewer #2:

We thank Reviewer 2 for highly evaluating our work and recommending the publication in Nature Communications. We have now answered the comments from the Reviewer.

One comment regarding the asymmetry between xz and yz after pumping in the weak fluence regime (Fig. 3a and b). It seems that what the authors put in the response to be rather important, that there is an asymmetric scattering channel for xz and yz between Γ and M in the nematic state, which could be the reason for the missing yz spectral weight at Γ after pumping. At least this seems consistent with the equilibrium ARPES results (and perhaps even the STM result) on FeSe in that the electron pocket carrying xz at M has been missing while the observed electron pocket is dominated by yz . Could it be useful to include a sentence in the main text pointing out this difference of behavior between xz and yz , if not the potential

interpretation, at least a statement of the experimental observation?

We thank Reviewer 2 for his/her valuable suggestion. We added the sentences regarding the origin of the asymmetry between xz and yz after pumping in the weak fluence regime to the new reference #29 in line 22, page 4, as follows.

“[29] Substantial spectral weight depletion in the yz states after pumping might be attributed to the photo-excited yz electrons partly trapped at the M point. Because of the semi-metallic electronic structure, some part of electrons excited by 1.5 eV photons at Γ may quickly gather around the electron bands at M. The momentum-dependent sign-inversion of orbital polarization (Ref.17) realizes the yz dominated electron pocket near E_F at M, which may work as the reservoir for photoexcited yz electrons. To fully understand these dynamics, the TARPES covering the whole Brillouin zone is desired.”

Reply to the Reviewer #3:

We thank Reviewer 3 for highly evaluating our work and giving us valuable comments. We have now answered all the comments from the Reviewer.

However the final discussion of the article is too speculative. I support publication in Nature Comm. but I encourage the authors fix the following issues:

a) It would be good to state that the $1/|T-T_c|$ behavior is expected from Gaussian fluctuations and not does not provide information about the universality class and critical exponents.

We agree with the Reviewer's statement and revised the description as the following.

We revised the statement in line 22, page 6 as follows.

“the critical T -linear behavior was found in the inverse of the QEP intensity above T_s (Ref. 15).”

→

“the T -linear behavior was found in the inverse of the QEP intensity above T_s (Ref. 15), indicative of the Gaussian fluctuation evolving in this regime.”

b) The dynamical scaling observed when the system is weakly perturbed and the temperature approach T_c is conceptually different the one observed when approaching the fluence to a threshold value. The authors may rather consider non-equilibrium criticality to the explain their data.

We agree that the present comparison between the TARPES and electronic Raman scattering is somewhat speculative, and more detailed discussion on non-equilibrium critical phenomenon is certainly necessary. Nevertheless, the theoretical basis for such discussion is still severely lacking, and developing it may be beyond the scope of this paper. We strongly hope our work will stimulate the future investigations.

According to the Reviewer’s advice, we added some discussion to stress the above point (line 20, page 7 in the present manuscript) .

We added the discussion (line 20, page 7 in the present manuscript):

“This behavior is seemingly related to the critical nematic fluctuation, nevertheless, future theoretical studies on non-equilibrium critical phenomena are highly necessary.”

c) On the base of the data that I have seen, there is no reason to claim that the excitation of nematic order is decoupled to the lattice. On one hand, the conjecture of an underlying electronic transition at lower temperature is badly posed. Indeed, it is the relevant coupling to the lattice that defines the experimetal T_c . On the other hand, the observed critical fluence is not particularly small and also corroborates a significant coupling to the lattice. As a consequence the authors should revise misleading statements as "These real-time observations reveal the nature electronic nematic excitation decoupled from the underlying lattice."

We thank for the Reviewer’s valuable comments. We agree that the experimental T_c (i.e. $T_s = 90$ K) in this case mostly reflects the degree of coupling to the lattice, and the discussion on (de)coupling of the electronic nematicity and the lattice should be provided with most careful

considerations. In the previous manuscript, our description on this issue had been perhaps too assertive.

Regarding the electronic nematicity in Fe-based superconductors, there have been big efforts to experimentally decouple it from the effect of underlying orthorhombic lattice. Starting from the seminal work of Chu et al. [Science **337**, 710 (2012)], the elastoresistivity measurements have revealed the Curie-Weiss like enhancement of the anisotropic response at $T > T_s$ [Science **352**, 958 (2016), PNAS **113**, 8139 (2016), Phys. Rev. B **96**, 205133 (2017) etc.]. Based on the Landau model assuming the coupling of uniaxial strain and electronic nematicity, they derived that the nematic susceptibility probed by elastoresistivity is dominated by purely electronic origin, and the mean-field transition temperature should correspond to the electronic transition temperature [Science **337**, 710 (2012)]. On the other hand, through the electronic Raman scattering studies [Phys. Rev. B **82**, 180521(R) (2010), Phys. Rev. Lett. **111**, 267001 (2013), PNAS **113**, 9177 (2016), arXiv:1710.09892 etc.], the enhancement of the low-energy response in B_{1g} symmetry has been theoretically shown to be unaffected by coupling to the lattice strain and reflect the pure electronic nematicity [Gallais et al., C. R. Physique **17**, 113 (2016)]. In these studies, the mean-field transition temperature of the Curie-Weiss-like nematic susceptibility is also understood as the electronic transition temperature. There could be some remaining debate on the strict theoretical validity of the models and/or the accuracy of electronic (mean-field) transition temperature estimated by the extrapolation at $T > T_s$. However, now that extensive discussions are made through these publications in the Fe-superconductor community, we think that it is worth noting and comparing them with our present result. To more carefully describe these issues, we modified the text as follows.

We modified the text (line 22, page 6):

“In the XY Raman spectrum, the T -linear behavior was found in the inverse of the QEP intensity above T_s (Ref. 15), indicative of the Gaussian fluctuation evolving in this regime. Similarly, the elastoresistivity measurement had also revealed the existence of electronic nematic fluctuation at $T > T_s$ interpreted as the Curie-Weiss-like nematic susceptibility.³¹ Through the analysis of the T -dependent nematic susceptibility in the form of $|T - T_0|^{-1}$, the authors discuss the mean-field transition temperature T_0 in terms of the ideal nematic transition purely driven by electrons without any influence of lattice.^{15,31,}”

Regarding our present TARPES data, we discuss the F -dependence in analogy with the T -dependent critical behavior. One of the “threshold” $F_c = 220 \mu\text{Jcm}^{-2}$, where the completely circular Fermi surface is attained at 120 fs, should naively correspond to the experimental T_c (90 K). “Another threshold” we are considering here is the F ($< 30 \mu\text{Jcm}^{-2}$, as explained in line 8, page 7)

where the finite dissolution of nematicity starts to be observed in the ultrafast regime (Fig. 2c). By the pump fluence of $F \sim 30 \mu\text{Jcm}^{-2}$, the electron temperature should only rise up to 45 K, where the change of Fermi surface nematicity is negligible in the equilibrium state. This tells us that *in the ultrafast regime* (~ 120 fs), the Fermi surface nematicity is seemingly decoupled from the lattice. We also note that according to the two-temperature model, the electron temperature is strongly deviated from the lattice temperature *in the ultrafast regime* (< 1 ps). To make this point clearer, we revised the text as follows.

We revised the text in the abstract:

“These real-time observations reveal the nature of the electronic nematic excitation decoupled from the underlying lattice.”

→

“These real-time observations reveal the nature of the electronic nematic excitation instantly decoupled from the underlying lattice.”

We also added the word “instantly” in line 18, page 7 in the present manuscript.

Summary of changes

1. We revised the sentence in the abstract as follows.

“These real-time observations reveal the nature of the electronic nematic excitation decoupled from the underlying lattice.”

→

“These real-time observations reveal the nature of the electronic nematic excitation instantly decoupled from the underlying lattice.”

2. We removed the sentence below in line 5, page 4 in the previous manuscript.

Such an oscillatory response of nematic FS for $F > F_c$ is not likely attributed to the usual coherent phonons^{24-26,28}, since the energy of optical phonons in FeSe (~ 22 meV) is much higher than the observed oscillation (3.1 meV for $220 \mu\text{Jcm}^{-2}$).

3. We added some description on A_{1g} phonons inline 9, page 4.

“The time scale of the oscillatory response (1.4 ps, 3.1 meV) is considerably slow as compared to the coherent A_{1g} optical phonon (190 fs, 22 meV) which is known to strongly couple to the electronic state in this system²⁴⁻²⁶. Their potential interplay is unfortunately hidden in the present TARPES data, possibly due to the duration of the pump pulse (170 fs) comparable to the time period of A_{1g} mode (190 fs) that tends to vanish the coherent oscillation.”

4. We modified the sentences below in line 8, page 6 in the present manuscript as follows.

The nematic-orbital excitation obtained in the present TARPES shows striking similarities with the nematic dynamics in thermal equilibrium as probed by the recent Raman scattering measurements.

→

Now we discuss the nematic-orbital excitation obtained in the present TARPES by comparing with the nematic dynamics in thermal equilibrium as probed by the recent Raman scattering measurements.

5. We removed the sentences below in line 14, page 6 in the previous manuscript.

The XYRaman spectrum further reveals a peak at 3.6 meV in the superconducting state ($T < T_c$), which is interpreted in Ref [30] as the nematic resonance mode acquiring the coherence by the superconducting gap opening. This energy scale is fairly close to that of the damped k_F oscillation (3.1 meV) observed by TARPES near F_c , thus suggesting the similarity in its origin. With these facts, we presently consider that the nematic-orbital excitation obtained by TARPES should be associated with the QEP (i.e. nematic fluctuation) in the Raman study.

6. We revised the statement in line 22, page 6 as follows.

“In the *XY* Raman spectrum, the critical *T*-linear behavior was found in the inverse of the QEP intensity above T_s (Ref. 15). By the detailed analysis of the Curie-Weiss-like *T*-dependent nematic susceptibility in the form of $|T - T_0|^{-1}$,¹⁵ the authors derived the bare electronic nematic transition temperature T_0 that should describe the ideal nematic transition purely driven by electrons without any influence of lattice.”

→

“In the *XY* Raman spectrum, the *T*-linear behavior was found in the inverse of the QEP intensity above T_s (Ref. 15), indicative of the Gaussian fluctuation evolving in this regime. Similarly, the elastoresistivity measurement had also revealed the existence of electronic nematic fluctuation at $T > T_s$ interpreted as the Curie-Weiss-like nematic susceptibility.³¹ Through the analysis of the *T*-dependent nematic susceptibility in the form of $|T - T_0|^{-1}$, the authors discuss the mean-field transition temperature T_0 in terms of the ideal nematic transition purely driven by electrons without any influence of lattice.^{15,31}”

7. We added the word “instantly” in line 18, page 7 in the present manuscript.
8. We revised the sentences in line 20, page 7 in the present manuscript as follows.

which is seemingly related to the critical nematic fluctuation

→

This behavior is seemingly related to the critical nematic fluctuation, nevertheless, future theoretical studies on non-equilibrium critical phenomena are highly necessary.

9. We added the word “Experimentally,” in line 24, page 7 in the present manuscript.
10. We added the sentence below in line 2, page 8 in the previous manuscript.

Systematic time-resolved diffraction measurements will also help understanding the possible interplay among the nematic excitation and the optical / acoustic phonons.^{25, 34}

11. The previous reference #28 was removed.
12. We added the sentences to new reference #29 in line 22, page 4, as follows

[29] Substantial spectral weight depletion in the *yz* states after pumping might be attributed to the photo-excited *yz* electrons which are partly trapped at the M point. Because of the semi-metallic

electronic structure, some part of electrons excited by 1.5 eV photons at Γ may quickly gather around the electron bands at M. The momentum-dependent sign-inversion of orbital polarization (Ref. 17) realizes the yz dominated electron pocket near E_F at M, which may work as the reservoir for photoexcited yz electrons. To fully understand these dynamics, the TARPES covering the whole Brillouin zone is desired.

13. We added the reference #34

[34] Nakamura, A. *et al.*, *Evaluation of photo-induced shear strain in monoclinic VTe₂ by ultrafast electron diffraction*. *Appl. Phys. Exp.* **11**, 092601 (2018).

14. We revised the word “FS” into “Fermi surface” throughout the manuscript.

15. We added the words “of $F = 220 \mu\text{Jcm}^{-2}$ ” in the caption of Fig. 2a,b.

16. We added the zero label and y-axis scale in the data in Fig. 2d.

REVIEWERS' COMMENTS:

Reviewer #3 (Remarks to the Author):

The authors did take in account all my remarks.
I am satisfied by their reply and I support publication in Nature Communication.

Reviewer #1 (Remarks to the Author):

The authors have answered my question in a satisfactory way. I suggest to publish the paper in its current form in Nature Communications.